# Ensemble of nucleic acid absolute quantitation modules for copy number variation detection and RNA profiling

Lucia Ruojia Wu [1,2,6], Peng Dai [1,3,6], Michael Xiangjiang Wang [1], Sherry Xi Chen[1,3], Evan N. Cohen [4], Gitanjali Jayachandran[4], Jinny Xuemeng Zhang[3], Angela V. Serrano[3], Nina Guanyi Xie[1], Naoto T. Ueno [5], James M. Reuben[4], Carlos H. Barcenas[5✉] & David Yu Zhang[3✉]

Current gold standard for absolute quantitation of a specific DNA sequence is droplet digital PCR (ddPCR), which has been applied to copy number variation (CNV) detection. However, the number of quantitation modules in ddPCR is limited by fluorescence channels, which thus limits the CNV sensitivity due to sampling error following Poisson distribution. Here we develop a PCR-based molecular barcoding NGS approach, quantitative amplicon sequencing (QASeq), for accurate absolute quantitation scalable to over 200 quantitation modules. By attaching barcodes to individual target molecules with high efficiency, 2-plex QASeq exhibits higher and more consistent conversion yield than ddPCR in absolute molecule count quantitation. Multiplexed QASeq improves CNV sensitivity allowing confident distinguishment of 2.05 ploidy from normal 2.00 ploidy. We apply multiplexed QASeq to serial longitudinal plasma cfDNA samples from patients with metastatic *ERBB2*+ (*HER2*+) breast cancer seeking association with tumor progression. We further show an RNA QASeq panel for targeted expression profiling.

[1] Department of Bioengineering, Rice University, Houston, TX, USA. [2] School of Pharmaceutical Sciences, Capital Medical University, Beijing, China. [3] NuProbe USA, Houston, TX, USA. [4] Department of Hematopathology, The University of Texas MD Anderson Cancer Center, Houston, TX, USA. [5] Department of Breast Medical Oncology, The University of Texas MD Anderson Cancer Center, Houston, TX, USA. [6] These authors contributed equally: Lucia Ruojia Wu, Peng Dai. ✉email: CHBarcenas@mdanderson.org; genomic.dave@gmail.com

Quantitation of specific nucleic acid sequences is the basis of many important biological applications; one example is gene ploidy calculation for detection of copy number variations (CNVs)[1–3], which is one of the most frequently observed genetic biomarker types in cancer[4,5]. It is present in 3–98% of tumor cases depending on the cancer type[6], and is clinically relevant as prognostic markers and as therapeutic targets[7–9]. Additionally, quantitation of specific RNA molecules for gene expression patterns profiling reflects the state of a cell or tissue[10,11] and may reveal pathological mechanisms underlying diseases[12–14].

Droplet digital PCR (ddPCR) is the gold standard for absolute quantitation of specific nucleic acid sequences[15,16]. The quantitation precision enabled small fold change measurements in CNV detection. The minimum copy number gain that can be distinguished from normal ploidy of 2.0 was improved from 3.0 using quantitative PCR (qPCR) to approximately 2.4 using ddPCR[17], rendering ddPCR useful for CNV detection in clinical settings[18]. However, improved CNV sensitivity is still highly desired especially for cell-free DNA (cfDNA) samples in which tumor DNA (ctDNA) are significantly "diluted" by DNA from normal tissues[19,20]. The intrinsic limitation due to stochasticity in molecule sampling process leads to the observed number of DNA molecules, and thus the observed ploidy, deviating from the expected "true value" in CNV quantitation (Supplementary Fig. S1). Poisson statistics can be used to model this sampling process: the standard deviation of a Poisson variable $X$ is $\sqrt{X}$, and the coefficient of variation (CV) is $\frac{1}{\sqrt{X}}$. Thus, in principle, increasing the DNA input amount or the number of genomic sites to quantify in the same gene would improve the limit of detection (LoD). Because the DNA input is usually limited especially in plasma-derived cfDNA, to overcome the Poisson distribution problem, ensemble of quantitation modules to sample a large number of independent genomic loci on the same gene is required to further improve CNV sensitivity. Highly multiplexed ddPCR remains challenging due to limited fluorescence channels.

As an alternative approach to ddPCR, we present Quantitative Amplicon Sequencing (QASeq), a PCR-based molecular barcoding NGS approach for accurate absolute quantitation which is compatible with high multiplexing. Herein, we demonstrate that 2-plex QASeq exhibited higher and more consistent conversion yield than ddPCR in absolute molecule count quantitation, and enables CNV quantitation accuracy similar to ddPCR. Multiplexed QASeq improves LoD to allow confident distinguishing 2.05 ploidy from normal 2.00 ploidy, and is applied to longitudinal serial 57 plasma cfDNA samples from patients with metastatic *ERBB2+* (*HER2+*) breast cancer. Finally, an RNA QASeq panel covering 20 genes are demonstrated on a wide range of RNA samples including tumor and placenta FFPE RNA.

## Results

### QASeq development and 2-plex demonstration.
Unique molecular identifiers (UMIs)[21–24] are attached to individual input DNA strand via two cycles of PCR with long annealing time for high and uniform barcoding efficiency. After further amplification, NGS reads originating from the same input DNA strand carry the same UMI sequence and thus belong to the same UMI family. Therefore, the unique UMI family count represents the number of input DNA strands (Fig. 1a).

We demonstrated QASeq for absolute quantitation and copy number calculation using a 2-plex panel containing 2 quantitation modules in gene *ERBB2* (target) and *EIF2C1* (reference) respectively (Supplementary Note 2 and Note 7), and compared with ddPCR for the same 2 genes side-by-side using five

replicated experiments for both methods (Fig. 1b). QASeq exhibited higher and more consistent conversion yield than ddPCR, where conversion yield is the fraction of input molecules that are observed in the experiment. 10 ng Human PBMC gDNA from the same healthy donor was used per experiment, corresponding to 2,790 haploid copies. QASeq showed higher conversion yield (86% on average) than ddPCR (53% on average). The coefficient of variation (CV) of molecule count was lower for QASeq (5.0% for *ERBB2*, 2.5% for *EIF2C1*) than for ddPCR (12.8% for *ERBB2*, 13.3% for *EIF2C1*) in 5 replicates.

High dynamic range of DNA input was observed for absolute quantitation using 2-plex QASeq (Fig. 1c). Observed *ERBB2* molecule counts by QASeq were close to the expected value calculated from DNA input amount because of high conversion yield. Lower conversion yield at 1 ng input DNA was possibly a result of material loss at low concentration.

*ERBB2* ploidy calculated from QASeq was accurate and highly reproducible (Fig. 1d). *ERBB2* ploidy was calculated as 2 × *ERBB2* molecule counts / *EIF2C1* molecule counts. The mean of five replicates was 1.98, which is close to normal ploidy of 2.00. The CV for calculated *ERBB2* ploidy was 5% in five replicates. Spike-in cell-line DNA samples with different expected *ERBB2* ploidy were assayed by 2-plex QASeq and ddPCR, and high correlation in calculated ploidy was observed between the methods (Supplementary Fig. S2).

The histogram for observed UMI family size distribution followed log-normal distribution after removal of small families (Fig. 1e), because in theory random yield difference in PCR efficiency was amplified exponentially in PCR cycles post UMI attachment. Large number of UMI families with UMI family size <3 were observed. These small UMI families were not used for molecule counting as they are likely results of polymerase and sequencing errors.

### Highly multiplexed QASeq for CNV detection.
Combination of multiple absolute quantitation modules for CNV detection was demonstrated to overcome the stochasticity in sampling. Experimentally, we did observe that the stochastic error in copy number quantitation reduced as a function of number of quantitation modules in the gene (Fig. 2a). A QASeq panel with 175 modules was designed for *ERBB2* CNV detection, with 49 quantitation modules in *ERBB2* and 123 modules in other regions of human genome serving as the reference (Supplementary Note 3). The rest of three modules are in Chromosome X thus are not used in CNV analysis. Five technical replicates were conducted with 8.3 ng healthy PBMC gDNA per library. 1 to 49 modules were used for *ERBB2* ploidy quantitation. The CV of *ERBB2* ploidy in five replicated experiments was reduced from 3.25% to 0.69% as the number of modules increases, consistent with theory based on Poisson distribution.

Combination of multiple quantitation modules in QASeq allowed confident discrimination between 2.05 and 2.00 ploidy *ERBB2* samples using QASeq (Fig. 2b). 2.05 ploidy sample was prepared by mixing a normal PBMC DNA sample and *ERBB2*-positive cell-line (SK-BR-3) DNA. Normal sample was tested in quadruplicates, and 2.05 ploidy sample was tested in duplicates.

### Multiplexed QASeq for CNV in tumor tissue samples.
QASeq was applied to 18 fresh/frozen (FF) tumor samples from 16 breast cancer patients and was compared with both ddPCR and immunohistochemistry (IHC) results on *ERBB2*. QASeq *ERBB2* ploidy results were concordant with IHC and ddPCR from the tumor tissue (Fig. 2c, d) with potentially fewer false positives. *ERBB2* ploidy from QASeq showed high correlation with ddPCR (Supplementary Fig. S3). For the single sample with discordance

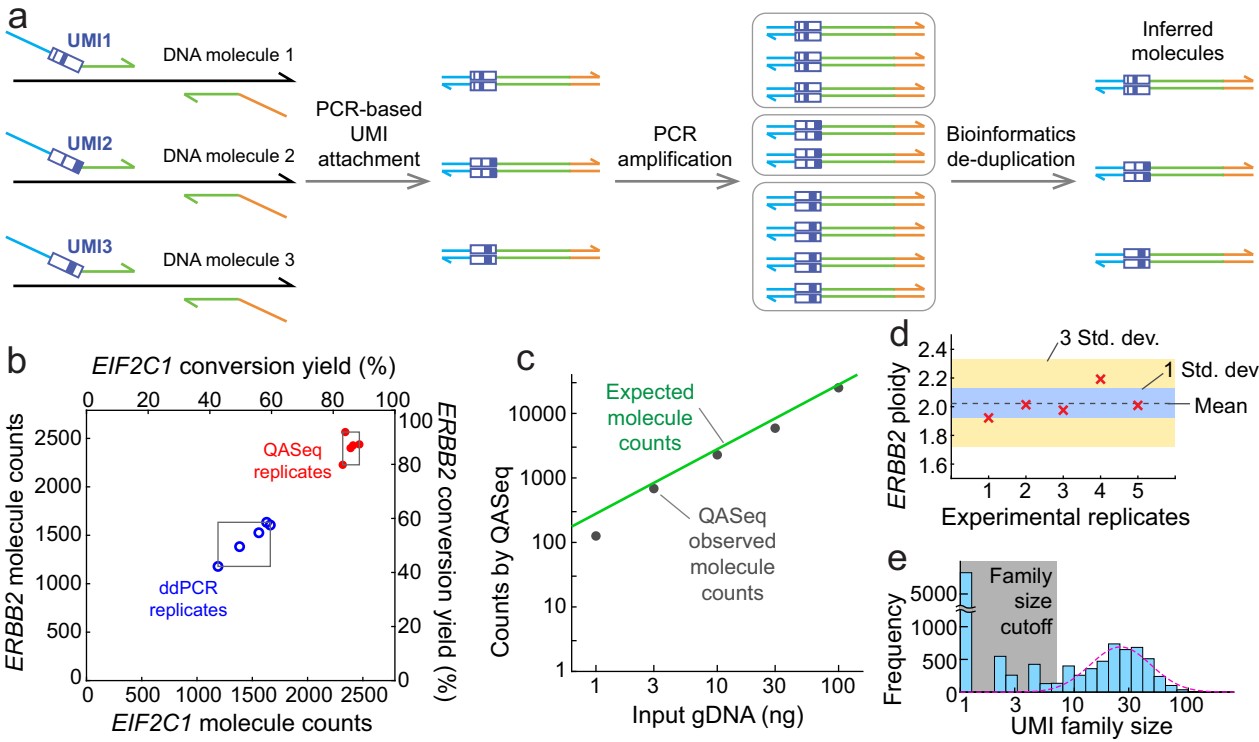

**Fig. 1 NGS-based QASeq DNA absolute quantitation modules. a** Schematic of a single QASeq module for quantitation of DNA bearing a specific nucleotide sequence. Each input DNA strand was attached with a UMI by 2 PCR cycles. Further PCR amplification was performed after removing non-extended primers bearing UMI. NGS reads originating from the same input DNA strand carry the same UMI sequence, thus unique UMI family count represents the number of input DNA strands. **b** Quantitation of *ERBB2* and *EIF2C1* molecule count using 2-plex QASeq and ddPCR. Five replicated experiments were performed for both methods with 10 ng gDNA input. QASeq showed higher conversion yield (86% on average) than ddPCR (53% on average). The CV of molecule count was lower for QASeq (5.0% for *ERBB2*, 2.5% for *EIF2C1*) than for ddPCR (12.8% for *ERBB2*, 13.3% for *EIF2C1*). **c** Absolute quantitation of different DNA input. Observed *ERBB2* molecule counts by QASeq were close to expected molecule counts calculated from DNA input amount because of high conversion yield. Lower conversion yield at 1 ng input DNA was possibly a result of material loss at low concentration. **d** Technical variation of *ERBB2* ploidy using 2-plex QASeq. The mean of 5 replicates was 1.98, which is close to normal ploidy of 2. **e** UMI family size distribution and data processing. UMI family size follows log-normal distribution after removing small families. Family size cutoff was calculated as 5% of the mean of top 3 largest family size here.

between QASeq and IHC, ddPCR agreed with QASeq. For all the three samples with discordance between QASeq and ddPCR, ddPCR ploidy were between 2.5 and 3 and IHC agreed with QASeq for negative call. There was no case where IHC and ddPCR agreed on a call that conflicted with QASeq results.

175-plex QASeq is theoretically equivalent to $C(175, 2) = 15225$ different ddPCR CNV assays (Supplementary Fig. S4). To utilize quantitation modules beyond just calculating *ERBB2* ploidy, modules in 'reference' were further grouped based on gene. Ploidy for 10 genes and 2 chromosomal regions were calculated from QASeq. To reduce the false positives of CNV calls in clinical samples and account for potential poor sample quality, sequential Mann–Whitney *U* tests on each gene of interest were performed (Supplementary Fig. S5, see Supplementary Note 3 on data analysis for multiplexed QASeq with >2 quantitation modules). As an example, ploidy of each of the 175 quantitation modules for a normal PBMC DNA sample and for an FF DNA sample from breast cancer tumor section was shown in Fig. 2e. In clinical sample analysis, the ploidy values for a gene will be reported as 2.00 if there is no statistical difference between gene of interest and reference by Mann–Whitney *U* test. We summarized the CNV results in all of 18 FF samples (Fig. 2f). The LoD for *ERBB2* CNV was calculated from the five technical replicates in healthy gDNA to be 1.97 ploidy for copy number loss and 2.04 for gain (Supplementary Table S2 and Supplementary Note 3). Additionally, *ERBB2* ploidy in PBMC DNA from 10 different

healthy donors was assessed with 175-plex QASeq and ddPCR respectively for biological variability (Supplementary Fig. S6). There is no sample with ploidy deviating from 2.00 for over 10%, but ddPCR showed wider ploidy range (1.8–2.1) than QASeq (1.9–2.0) in the 10 normal samples.

QASeq could improve clinical sensitivity in CNV assessment. The ploidy values of all observed gene ploidies in 18 tumor DNA samples are plotted as a histogram (Supplementary Fig. S7). Using methods with LoD at 1.6 and 2.4 ploidy, such as ddPCR, 44% of the CNVs may be missed. Additionally, QASeq allows quantitation of mutations down to 0.1% variant allele frequency (VAF) with UMI error correction (Supplementary Fig. S8). As an NGS-based quantitation method, sequence mutation calling is performed in addition to the copy number calculation. We designed QASeq panel amplicons to include hot spot mutations commonly observed in breast cancer (Supplementary Note 4).

**Study of QASeq liquid biopsy results with disease progression in *ERBB2*+ (*HER2*+) metastatic breast cancer patients**. QASeq breast cancer liquid biopsy panel (Supplementary Note 7) was applied to serial longitudinal plasma cell-free DNA (cfDNA) samples and was compared with disease progression. 57 plasma cfDNA samples from 15 patients with *ERBB2*+ metastatic breast cancer were tested by QASeq, with 2–8 samples per patient, where all patients had a baseline sample obtained at the moment of diagnosis of metastatic breast cancer, and follow-up samples

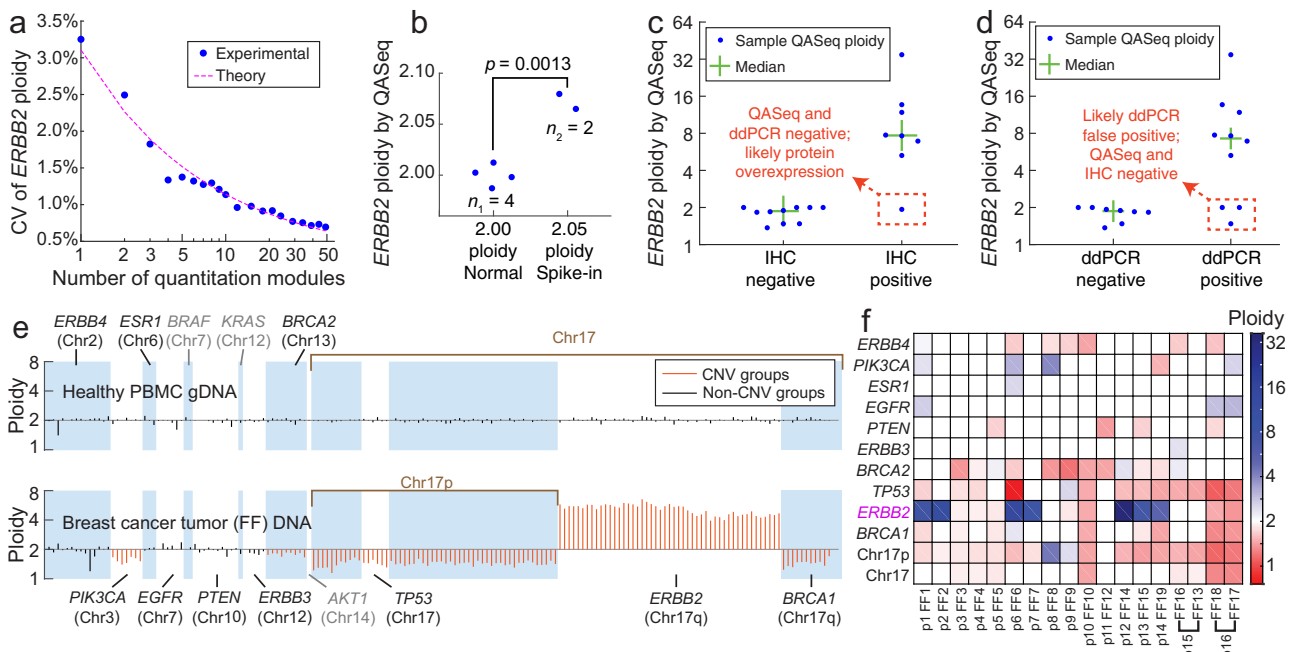

**Fig. 2 Combination of multiple absolute quantitation modules for CNV detection. a** Stochastic error in copy number quantitation was reduced by increasing the number of quantitation modules in the gene. *ERBB2* ploidy was calculated as 2 times the ratio between the mean of UMI family counts from modules in *ERBB2* and in the reference. Because there are multiple possibilities of module down-selection, here each datapoint represents the average of 30 randomized selections. 123 modules served as the reference for ploidy calculation. The CV of 1 module was lower than that in Fig. 1d, because only 1 module was used for both the reference and *ERBB2*. **b** Discriminating 2.05 and 2.00 ploidy *ERBB2* samples using QASeq. The *P*-value calculated using the two-sided *t*-test was 0.0013. 2.00 ploidy sample was a normal PBMC DNA sample (ZB4) tested in four independent experiments; 2.05 ploidy sample was prepared by mixing the same normal PBMC DNA sample (ZB4) and ERBB2-positive cell-line (SK-BR-3) DNA, and was tested in two independent experiments. **c** Concordance of QASeq *ERBB2* ploidy with IHC. For the sample with discordance between QASeq and IHC, ddPCR agreed with QASeq; this might be a result of *ERBB2* over-expression at protein level due to changes in promoters or enhancers. **d** Concordance of QASeq *ERBB2* ploidy with ddPCR. For the three samples with discordance between QASeq and ddPCR, IHC agreed with QASeq. **e** Ploidy of each QASeq quantitation module for a healthy PBMC donor (top) and for a breast cancer fresh/frozen (FF) tumor section (bottom). Genes with called CNVs are displayed in orange, genes within expected variation are shown in black. The genes in gray have <3 modules per gene in QASeq, and thus cannot be used to calculate gene ploidy using Mann–Whitney *U* test. **f** QASeq CNV calls for 18 fresh/frozen breast tumor sections. Two non-adjacent tumor sections were tested from patients p15 and p16. Between 5 and 8 ng DNA input was used based on availability.

were obtained at different time points for each patient. All patients had a diagnosis of *ERBB2*+ (IHC II+ or III+ or FISH+) metastatic breast cancer and the *ERBB2* status was confirmed from a biopsy obtained from tumor tissue.

We summarized *ERBB2* ploidy change in cfDNA and disease progression dates for each patient using a swimmer plot (Fig. 3a). There were 8 patients who developed disease progression who had a plasma sample collected within 6 months before or after the disease progression. *ERBB2* amplification or increase of *ERBB2* ploidy relative to the previous time point was observed in 6 out of the 8 patients who developed disease progression. In the other two patients (de-identified patient ID 2697 and 2366), disease progression was reported 4 times for each patient and abnormal *ERBB2* CNV can only explain part of the disease progression. Significant allele frequency changes in *PIK3CA* G1049R mutation in patient 2697 and in SNP rs1309838194 in patient 2366 were correlated with disease progression respectively (Fig. 3b). *PIK3CA* G1049R mutation (COSV55874453) is considered to be a structural damaging alteration as disease-causing drivers[25,26] in breast cancer. The VAF for *PIK3CA* G1049R in circulating cfDNA was increased by over 10-fold during follow-up, serving as evidence for increased tumor fraction. The *PIK3CA* mutation may have contributed to disease progression. In patient 2366, VAF for the SNP rs1309838194 in *ERBB2* changed from around 50% (heterozygous at baseline time point) to 80% during follow-up, which indicated increased tumor-derived DNA in plasma and

may be associated with disease progression. Since the overall *ERBB2* ploidy in plasma was still normal, we think that the copy-neutral loss of heterozygosity (LOH)[27,28] may be present in the tumor, leading to SNP allelic imbalance. Taken the CNV and mutation results together, abnormal change was reported by QASeq in all 8 patients who developed disease progression.

The 15 cases were classified according to whether or not there were abnormal molecular findings in QASeq liquid biopsy and whether or not disease progression was clinically observed (Fig. 3c). Chi-square test suggested that the QASeq result and progression are not statistically independent ($p = 0.038$). Although all cases with disease progression were featured with QASeq abnormal findings showing good sensitivity, increase of *ERBB2* ploidy was also observed in 4 patients who did not develop disease progression within this time frame. Because all patients presented here were treated with *ERBB2*-targeted therapy trastuzumab and pertuzumab until disease progression, molecular change may not translate to clinically observed disease progression in all cases.

Furthermore, tumor *ERBB2* ploidy was inferred from plasma QASeq results and was compared with FISH. Because both CNV and mutation information are available from QASeq, tumor fraction in plasma cfDNA can be estimated based on VAF of tumor mutation; tumor ploidy can be calculated with plasma ploidy and tumor fraction. We demonstrated this normalization in two cases (Fig. 3d), where pathogenic mutation was observed

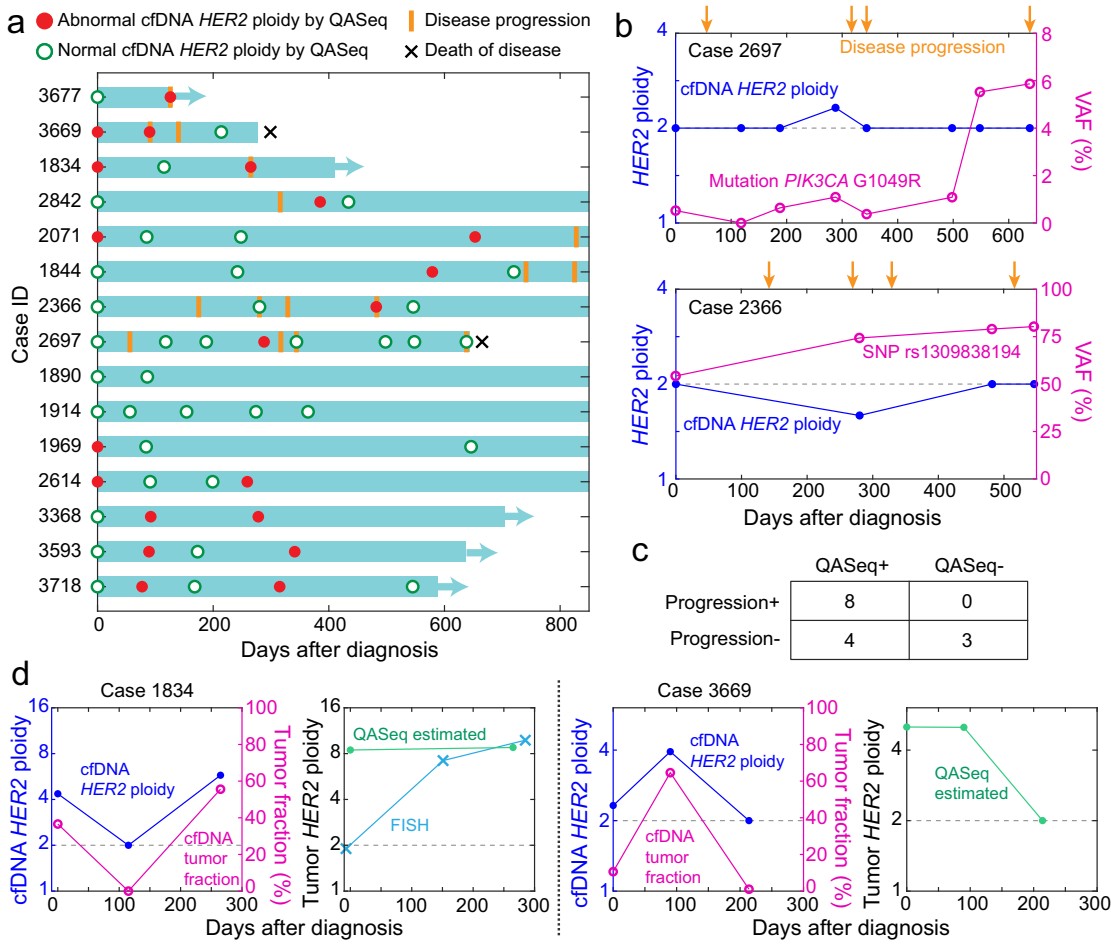

**Fig. 3 QASeq for longitudinal study of plasma samples from 15 *ERBB2*+ metastatic breast cancer patients. a** Swimmer plot of clinical course and molecular findings of patients. QASeq identified cfDNA *ERBB2* amplification or increase of *ERBB2* ploidy relative to the previous point in 6 out of the 8 patients with progression within 6 months of plasma sampling. The other 2 patients (sample ID 2697 and 2366) were each reported with 4 progressions. Although *ERBB2* amplification or increase of *ERBB2* ploidy was also observed in these two patients, abnormal *ERBB2* events are only associated with part of the progression. Between 5.6 and 8.3 ng DNA input was used based on availability. **b** *ERBB2* ploidy and mutation allele frequency change in plasma sample of patients 2697 and 2366. Significant allele frequency changes in *PIK3CA* G1049R mutation in patient 2697 and in SNP rs1309838194 in patient 2366 were correlated with progression. **c** Categorization of patients based on QASeq abnormal molecular findings and disease progression. Chi-square test suggested that the QASeq result and progression are not statistically independent (*p* = 0.038). **d** Plasma *ERBB2* ploidy normalization with tumor fraction to infer tumor *ERBB2* ploidy. Tumor FISH results at three time points were collected in patient 1834. QASeq detected *ERBB2* amplification in plasma 5 months earlier than FISH from tumor tissue. The inferred tumor *ERBB2* ploidy by QASeq was consistent with the available tumor ploidy from FISH. The inferred tumor *ERBB2* ploidy was generally stable in both of the two patients.

in baseline cfDNA with VAF between 1% and 30%. Tumor FISH results were collected at three time points in case 1834. QASeq detected *ERBB2* amplification in plasma cfDNA 5 months earlier than FISH from tumor tissue. In addition, the inferred tumor *ERBB2* ploidy by QASeq was consistent with the available tumor ploidy from FISH. The inferred tumor *ERBB2* ploidy was generally stable in both of the two patients, so *ERBB2* ploidy change in cfDNA was influenced by tumor fraction. Based on the correlation of QASeq results with progression and FISH results, we envision non-invasive and sensitive longitudinal study of CNV and mutation change in plasma by QASeq can help with understanding disease progression and resistance mechanism.

**RNA QASeq for gene expression level quantitation**. Next, we demonstrated QASeq technology for RNA quantitation in a variety of samples including tumor tissue FFPE RNA, total blood RNA and total liver RNA. The RNA sample was reverse transcribed to cDNA as input for QASeq. Random hexamer was

chosen as reverse transcription primer to be compatible with low-quality fragmented FFPE RNA. A targeted multigene breast cancer panel covering 78 amplicons in 15 cancer-related and 5 reference genes similar to Oncotype DX[29] panel was built (Supplementary Note 5). Expression of each gene is calculated from the molecule count of each amplicon, based on UMI count and conversion yield, and is further normalized relative to the expression level of the 5 reference genes in log2 scale (Supplementary Fig. S9).

The RNA quantitation accuracy was firstly validated using ERCC RNA spike-in mix. 16 ERCC sequences were targeted with 16 amplicons. The ERCC RNA sample was diluted and mixed with commercial human total liver RNA for a final expected molecule count between 3 and 100,000. The observed molecule count showed good correlation with the expected (Fig. 4a). QASeq quantitation for RNA was across five orders of magnitude and as few as 3 expected molecules were detected.

Reproducibility for expression level relative to reference genes was evaluated. Total liver RNA was assayed with breast cancer

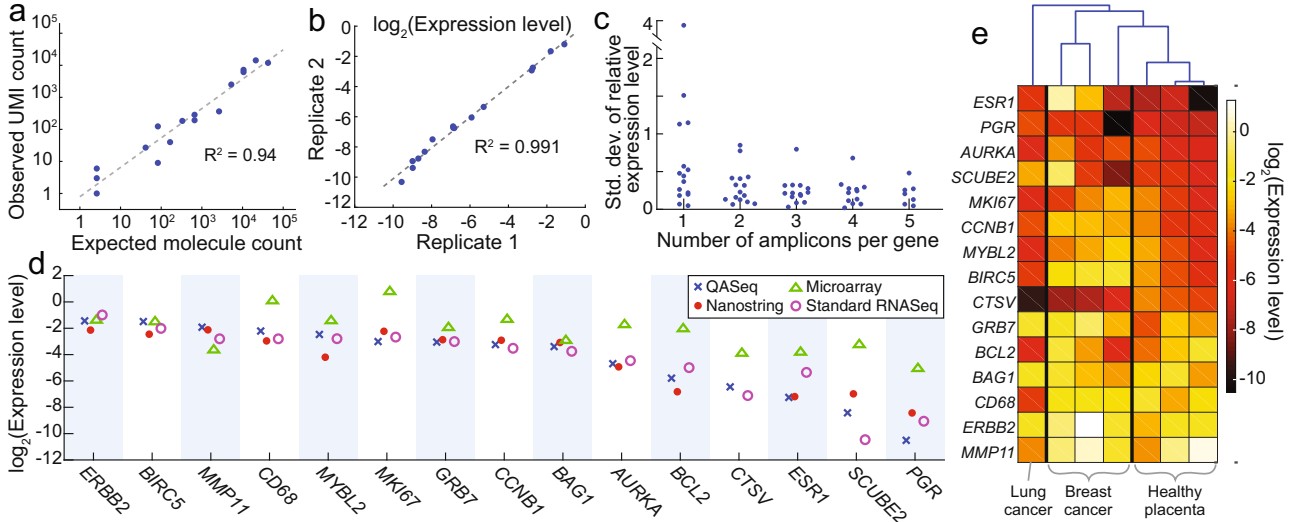

**Fig. 4 RNA QASeq for gene expression level quantitation. a** Quantitation accuracy validation with ERCC spike-in reference sample. **b** RNA QASeq quantitation reproducibility in technical replicates. 10 ng total liver RNA was used as input. **c** Multiple amplicons per gene reduced quantitation variability in technical triplicate analysis of total liver RNA sample. The standard deviation for relative expression level in triplicate experiments became lower as the number of amplicons per gene increased from 1 to 5. Only the amplicons at 5′ end of mRNA were used when analyzing a subset of amplicons. **d** RNA relative expression level quantitation side-by-side comparison using four different methods in one breast cancer FFPE RNA sample. **e** Relative expression level measured by RNA QASeq in four clinical FFPE tumor tissue samples and three normal placenta FFPE samples. Hierarchical clustering indicated the expression patterns were the most similar between normal placenta samples.

panel in replicates and the consistent expression level was observed (Fig. 4b). We observed that multiple quantitation modules (amplicons) per gene reduced quantitation variability in expression level. The standard deviation for relative expression level in triplicate experiments became lower as the number of amplicons per gene increased from 1 to 5 (Fig. 4c), with median standard deviation reduced from 0.44 to 0.21. The outlier was only observed when only 1 amplicon is considered.

RNA expression level from QASeq was extensively compared with other technologies including RNAseq[30], NanoString nCounter[31], Microarray[32], and RT-qPCR using FFPE RNA from breast cancer and lung cancer tissue. The expression level was normalized in the same way relative to the five reference genes for all the methods, and was summarized in Fig. 4d for a breast cancer FFPE RNA. RNA QASeq is consistent with RNAseq and NanoString nCounter. Microarray, however, showed poor correlation with any of the other methods. RNA QASeq was further compared with these technologies in a couple other samples (Supplementary Figs. S10–13). Nanostring showed high correlation with QASeq in all samples, but required much higher input amount than RNA QASeq. Low expression level species are dropped out at 10 ng as compared to the typical 150 ng input (Supplementary Fig. S14). Microarray showed poor concordance with both QASeq and nanostring in all samples (Supplementary Figs. S13, S15). QASeq was consistent with RNAseq in most samples. However, because RNAseq was a non-targeted approach, most reads were wasted on genes of no interest and coverage uniformity issue led to poor robustness for the quantitation of lowly expressed genes as it was observed in the two FFPE samples (Supplementary Fig. S11). RT-qPCR is consistent with UMI-based QASeq quantitation, but is limited by low multiplexing ability.

We summarized the relative expression level in four clinical FFPE and three normal placenta FFPE samples (Fig. 4d). Hierarchical clustering indicated the expression patterns were the most similar between normal placenta samples.

## Discussion

QASeq developed in this work provides an accurate absolute nucleic acid quantitation method that can be conveniently scaled up to high multiplexing, thus overcoming sampling error from Poisson distribution for CNV detection. We demonstrated confident distinguishment of 2.05 ploidy from 2.00 ploidy. Additional comparison between QASeq and other CNV calling tools including CovCopCan[33] and CNVKit[34] were performed in one normal PBMC DNA sample (expected *ERBB2* ploidy = 2.00), two reference spike-in samples prepared by mixing the normal PBMC DNA sample with *ERBB2*-positive cell-line (SK-BR-3) DNA (expected *ERBB2* ploidy = 2.05 and 2.20), and three clinical cfDNA samples from breast cancer patients with *ERBB2* amplification identified by QASeq (Ploidy = 2.17, 2.32 and 2.94). QASeq showed better sensitivity for *ERBB2* CNV detection. The cfDNA sample with *ERBB2* ploidy of 2.94 was called as *ERBB2* positive by all three methods; the sample with ploidy of 2.32 was identified by QASeq and CNVKit; the one with 2.17 ploidy was only identified by QASeq (Supplementary Note 6, Supplementary Table S4 and Supplementary Figs. S17–19). Absolute quantitation is based on a highly efficient PCR-based barcoding approach. Based on Poisson distribution theory and our observation, QASeq allows the construction of targeted panels with adjustable CNV sensitivity for each gene by changing the number of amplicons that cover each gene of interest. The recommended number of modules in gene of interest is dependent on the desired limit of detection for copy number variation detection. As shown in Fig. 2a, though stochastic error in copy number quantitation was reduced by increasing the number of quantitation modules in the gene, there is a diminishing marginal utility as the module number increases. As a reference for roughly estimating the number of modules for different LoD requirements, CNV LoD of different genes with different module numbers per gene in the 175-plex QASeq panel was summarized (Supplementary Table S2). Advances have been made to digital PCR systems in terms of both multiplexing[35,36] and dead volume. As an

alternative approach to ddPCR for nucleic acid quantitation, the multiplexing ability for QASeq is still over one order of magnitude higher than the best digital PCR leading to better CNV LoD using highly multiplexed QASeq panel.

Comparing to existing UMI-based methods which performed well on somatic mutation but were less sensitive to CNVs, the robust quantitation performance of QASeq is based on high and consistent conversion yield for each quantitation module, and the ability to scale up to high multiplexing. We chose to perform two cycles of PCR-based barcoding with long annealing time, because the fraction of DNA molecules in a sample represented in the final NGS library is low for ligation-based UMI attachment. As the number of primer pairs increases for a multiplex PCR amplification, there is a combinatorial explosion of potential primer dimers and non-specific genomic amplification. The dimer problem is more complicated when UMIs must be incorporated in the multiplex PCR. By combining Simulated Annealing Design using Dimer Likelihood Estimation (SADDLE)[37], a primer set optimization software developed in our lab, with a nested protocol, about 60% on-target rate is maintained even in the liquid biopsy panel with 223 modules. A 384-plex standard PCR panel showed >40% on-target rate after optimized by SADDLE algorithm[37]. Since a nested design is performed to further reduce non-specific amplification and primer dimer formation in QASeq protocol, we envision over 1,000 modules can be simultaneously tested by QASeq.

The most accessible sample type for cancer monitoring is cfDNA derived from plasma. However, CNV monitoring using cfDNA is understudied comparing to mutation detection[38]. Common CNV detection methods immunohistochemistry and in situ hybridization are limited to tissue/cell samples, and not applicable to non-invasive cfDNA. Other cfDNA-compatible methods including ddPCR, NGS panels such as FoundationOne and Guardant 360, and microarray are not sensitive enough, all requiring >25% heterozygous single copy loss or gain for detection which corresponds to 1.75 ploidy for loss and 2.25 ploidy for gain. The high CNV sensitivity of QASeq may allow better clinical sensitivity of copy number changes.

As a proof-of-concept demonstration, QASeq is used to infer tumor gene ploidy from cfDNA when tumor mutation is observed in two clinical cases. A single gene ploidy number in cfDNA may not be actionable because it is decided by both circulating tumor DNA (ctDNA) fraction and CNV in cancer cells, not to reflect any of the two explicitly. As an example, 1% tumor fraction in cfDNA with tumor copy number of 20, or 6% tumor fraction with tumor copy number of 5 has the same cfDNA overall ploidy (2.18), but the former may have better outcome to HER2-targeted therapy. Deconvolution of tumor CNV from cfDNA will be complicated when tumor tissue is highly heterogeneous. Clonal mutation, the mutation that is present in all cancer cells within tumors, needs to be identified in this case to infer the average gene ploidy in tumor.

We identified normal *ERBB2* ploidy but significant SNP allelic imbalance in *ERBB2* in one patient, which may be due to copy-neutral LOH in tumor. QASeq breast cancer panel covers hot spot cancer mutations while designing multiple regions in the gene of interest so not many SNPs are included. Inspired by this case study, we can intentionally add quantitation modules for SNP calling in the gene of interest to better identify allelic imbalance for higher confidence of LOH detection.

## Methods

**Ethical Statement**. The research complies with all relevant ethical regulations. All procedures performed in studies involving human participants were approved by Institutional Review Board at MD Anderson (protocols PA16-0507 and PA19-0375), and were in accordance with the 1964 Helsinki declaration and its later amendments or comparable ethical standards. Informed consent was obtained from all participants. Participants received no compensation, and the data was not used for any treatment decisions.

**Oligonucleotides and Reagents**. All oligonucleotides were purchased from Integrated DNA Technologies (100uM in IDTE, pH 8.0). Oligonucleotide sequences are provided in Supplementary Data 1 QASeq primer sequences. Primers in the 2-plex QASeq panel are dual-HPLC purified; primers in other panels are standard-desalted to reduce cost. Conversion yield can be slightly reduced using standard-desalted primer but the median conversion yield is still over 60%. Phusion High-Fidelity DNA polymerase and deoxynucleoside triphosphates (dNTPs) were purchased from New England Biolabs. PowerUp SYBR Green Master Mix was purchased from Thermo Fisher Scientific. iTaq Universal SYBR Green Supermix was purchased from Bio-Rad Laboratories. AMPure XP was purchased from Beckman Coulter. NGS index primers (NEBNext Multiplex Oligos for Illumina) were purchased from New England Biolabs.

**QASeq protocol**. Library preparation consists of three PCR reactions: UMI PCR, nested PCR and index PCR, all performed on a T100 Thermal Cycler (Bio-Rad). In UMI PCR, the DNA sample was mixed with 1U Phusion High-Fidelity DNA polymerase, Phusion HF buffer, forward and outer reverse primers (15 nM each), and dNTPs (0.2 mM each) to reach a total volume of 50 μL.

Thermal cycling started with 30 s at 98 °C, followed by two cycles of 10 s at 98 °C, 30 min at 63 °C and 15 s at 72 °C, and then two cycles of 10 s at 98 °C, 15 s at 63 °C and 15 s at 72 °C, finally five cycles of 10 s at 98 °C and 30 s at 71 °C. During the last 5 min of the second 30 min at 63 °C, 1.5 μM of each universal primer was added while keeping the reactions inside the thermal cycler. After UMI PCR, 1.6X AMPure XP beads purification was performed.

In nested PCR, the eluate from the previous step was mixed with PowerUp SYBR Green Master Mix (1X final concentration) and 15 nM each inner reverse primer. Thermal cycling started with 3 min at 95 °C, followed by 2 cycles of 10 s at 95 °C and 30 min at 60 °C. The PCR product was purified by 1.6X AMPure XP beads.

Next, index PCR was performed; the eluate from the previous step was mixed with iTaq Universal SYBR Green Supermix (1X final concentration) and 250 nM each NEBNext index primers. Thermal cycling started with a 3 min incubation step at 95 °C, followed by 25 cycles of 10 s at 95 °C and 30 s at 65 °C, and finally 2 min at 65 °C. After index PCR, double-side size selection (0.4X, 0.4X ratio) was performed. Libraries were normalized and loaded onto an Illumina sequencer.

DNA extracted from FF or blood samples was sheared to 150 bp peak length using Covaris LE220 Focused Ultrasonicator before library preparation.

In RNA QASeq, RNA sample was firstly reverse transcribed to cDNA as input for QASeq protocol. RNA was mixed with dNTP (0.5 mM), Murine RNase Inhibitor (8 U), M-MuLv buffer (1X), M-MuLV Reverse Transcriptase (8 U), and random hexamer (6 μM). The mixture was incubated at 25 °C for 5 min, at 42 °C for 60 min, and then inactivated at 65 °C for 20 min. The reaction mixture was directly used as input for UMI PCR without purification.

Sequencing was performed on HiSeq or NextSeq (Illumina) with 2 × 150 bp paired-end reads and dual 8 bp index. The recommended sequencing depth is 90,000X for 8.3 ng human DNA input (see Supplementary Information Note 6 for more details). QASeq replicates were performed by the same operator, using the same DNA input, with library preparation performed in the same day and sequenced in the same run.

**NGS data processing**. NGS adapter sequences were first removed from FASTQ data using custom Python code; alignment was performed using Bowtie2 software[39]. UMI grouping and CNV analysis were performed using custom Matlab code; a detailed description of the algorithm can be found in Supplementary Note 2 and Note 3. Mutation analysis was performed using custom Python and Matlab code; a detailed description can be found in Supplementary Note 4.

**Digital droplet PCR**. ddPCR CNV Assays from Bio-Rad were used in this study. Specifically, ddPCR Copy Number Assay: *ERBB2*, Human (Fluorophore: FAM, UniqueAssayID: dHsaCP1000116) and ddPCR Copy Number Assay: AGO1 (EIF2C1), Human (Fluorophore: HEX, UniqueAssayID: dHsaCP2500349) were purchased. Reaction setup, thermal cycling conditions and data acquisition were performed according to Bio-Rad protocol for ddPCR Copy Number Variation Assays. 10 ng of input DNA were used for each reaction. The ddPCR replicates were performed by the same operator, using the same DNA input, with droplet generation and PCR reaction performed in the same day and analyzed by Droplet Reader in the same run.

**Samples**. Fresh frozen (FF) breast tissue samples from breast cancer patients were purchased from OriGene Technologies. *ERBB2* status of the tumor tissue measured by immunohistochemistry (IHC) was obtained from the vendor. Genomic DNA from FF samples and buffy coat of blood samples was extracted using QIAamp DNA Mini (Qiagen) following the manufacturer's protocol.

56 plasma samples from 15 *ERBB2* + metastatic breast cancer patients in de-identified format were collected from MD Anderson Cancer Center. Patient Characteristics are summarized in Supplementary Note 7.

Cell-free DNA was extracted from plasma using QIAamp MinElute ccfDNA Mini Kit (Qiagen) following the manufacturer's protocol. Samples were quantified by qPCR with Human cell-line gDNA NA18537 as reference. The concentration calculated from qPCR reflects the amplifiable DNA.

Normal human placenta FFPE was purchased from BioChain. Total RNA from FFPE was extracted using RNeasy FFPE Kit (Qiagen). Human liver total RNA was purchased from Takara Bio. Human whole blood samples from healthy people were purchased from Zen-Bio Inc. RNA from fresh total blood was extracted using Monarch Total RNA Miniprep Kit (New England BioLabs).

*ERBB2*-positive cell-line (SK-BR-3) DNA was in the National Institute of Standards and Technology (NIST) Standard Reference Material 2373, and was purchased from ATCC (product name: SRM NIST-2373).

### Infer tumor *ERBB2* ploidy from plasma.

$$\text{Ploidy(cfDNA)} = \sum_1^i \text{Ploidy(tumor clone } i) \times \text{Fraction(tumor clone } i) \\ + 2.0 \times (1 - \sum_1^i \text{Fraction(tumor clone } i)) \tag{1}$$

$$\text{Mutation VAF(cfDNA)} = \sum_1^i \text{VAF(tumor clone } i) \times \text{Fraction(tumor clone } i) \tag{2}$$

Here Fraction(tumor clone $i$) is the fraction of circulation DNA derived from tumor subclone $i$ in cfDNA, Ploidy(tumor clone $i$) is the ERBB2 ploidy in pure tumor clone $i$, and VAF(tumor clone $i$) is the mutation VAF in pure tumor clone $i$.

Considering the complexity of tumor heterogeneity, here we proof-of-concept demonstrate the feasibility of inferring tumor ERBB2 ploidy from plasma, with two assumptions being made: (1) there is only one subclone in the tumor or we take the average of the tumor tissue to treat tumor as a whole, and (2) the VAF in that pure tumor is 50% (monoallelic).

When there is only one clone or we take the average of the tumor, equations are converted to:

$$\text{Ploidy(cfDNA)} = \text{Ploidy(tumor)} \times \text{Fraction(tumor)} + 2.0 \times (1 - \text{Fraction(tumor)}) \tag{3}$$

$$\text{Mutation VAF(cfDNA)} = \text{VAF(tumor)} \times \text{Fraction(tumor)} \tag{4}$$

We took pathogenic mutation observed in baseline cfDNA with VAF between 1% and 30% for tumor fraction calculation, to avoid the influence of SNP. Baseline mutation in 2 patients (1834 and 3669) were identified. We hypothesized mutation VAF in pure tumor is 50% (monoallelic), so that:

$$\text{Fraction(tumor)} = \text{Mutation VAF(cfDNA)} \times 2 \tag{5}$$

If the VAF in pure tumor is 100% (biallelic), tumor fraction calculated from Eq. (5) is 2-fold of the true value. Overall, we envision the inferred mean tumor ERBB2 ploidy based on Eqs. (3) and (5) should be within two-fold of the true value to help estimate tumor tissue information from plasma cfDNA.

**RNAseq**. Library preparation was performed using NEBNext Ultra II RNA Library Prep Kit for Illumina. Ribosomal RNA depletion was performed using NEBNext rRNA Depletion Kit v2. Raw fastq reads were initially quality filtered using Trimmomatic v0.39. Specifically, individual reads were trimmed to the longest continuous segment for which phred quality score ($Q$) was ≥20 ($Q \geq 20$ represents ~99% accuracy per nucleotide position). Reads shorter than 50 bp after trimming were discarded. Next, libraries were aligned to the human reference genome (GRCh38) using bowtie2 v2.4.4. After alignment, sam files were sorted and converted to bam files using samtools v1.12. HTseqv0.13.5 in mode 'intersection-strict' and with additional parameter '--minaqual 1' was used to estimate the number of reads that mapped to each gene of interest. Finally, StringTie v2.1.7 was used to calculate TPM-normalized gene abundance.

**Nanostring and microarray**. Extracted RNA samples were sent to Amsbio LLC for Nanostring test using nCounter Breast Cancer 360 V2 Panel, and were sent to UT Southwestern Medical Center Microarray Core Facility for GeneChip Human Transcriptome Array 2.0 (HTA 2.0) test.

**Reporting summary**. Further information on research design is available in the Nature Research Reporting Summary linked to this article.

## Data availability

The data supporting the results in this study are available within the paper and its Supplementary Information. The raw sequencing data and ddPCR data generated in this study have been deposited in NCBI Sequence Read Archive under BioProject ID PRJNA813699 and Figshare [https://doi.org/10.6084/m9.figshare.16529640]. Microarray data have been deposited in NCBI Gene Expression Omnibus under Accession ID GSE198252. Human reference genome (GRCh38) used in the study for alignment is accessed from NCBI under BioProject ID PRJNA31257. Source data are provided with this paper.

## Code availability

NGS data analysis pipeline for QASeq analysis is available from Github (https://github.com/wrj915/QASeq).

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

## Acknowledgements

This work was supported by NIH awards U01CA233364 and R01CA203964 to D.Y.Z., and CPRIT award RP180147 to D.Y.Z. The authors would like to acknowledge Franklin D Alvarez, Sanda Tin, and Anita L Wood for their help collecting and organizing patient samples.

## Author contributions

L.R.W. P.D., and D.Y.Z. conceived the project. L.R.W. and P.D. designed and conducted the experiments, and analyzed the data. M.X.W. performed experiments and analyzed the data. S.X.C. and N.G.X. analyzed the data. J.X.Z. and A.V.S. performed RNA QASeq experiments. E.N.C., G.J., N.T.U., J.M.R., and C.H.B. provided clinical plasma samples and analyzed the data. L.R.W., P.D., C.H.B., and D.Y.Z. wrote the paper with input from all authors.

## Competing interests

A US provisional patent application (No. 62/788,375) and an international patent application (No. PCT/US2020/012089) covering the use of QASeq technology have been filed in which the Rice University is the applicant, and D.Y.Z., L.R.W., P.D. are the inventors. P.D., L.R.W., S.X.C., and M.X.W. declares a competing interest in the form of consulting for NuProbe USA. D.Y.Z. is a co-founder and holds significant equity in NuProbe Global, Torus Biosystems, and Pana Bio. The remaining authors declare no competing interests.
