## [Peer Review File · Nature Communications]

REVIEWER COMMENTS

Reviewer #1 (Remarks to the Author):

The manuscript by Wu et al. describes an interesting new method for simultaneous absolute quantification of a certain number of targeted molecules based in next-generation sequencing. The authors describe the accuracy of the method compared to other techniques in different applications, such as estimation of CNV ploidy in tumor samples or cell free DNA and RNA quantification. They also show the application of the technique to different type of tumor samples and make a preliminary study of the association of disease progression and ERBB1 copy number in a small number of individuals. Overall the technique described appears to be reliable and accurate, improving some of the limitations of previous methods, and it is already included in the diagnostic company NUPROBE portfolio. However, so far the new biological insight gained is limited.

Major comments:

1. Given that the main contribution is the new method description, I think it would be useful for potential users to provide more practical information on its performance and diagnostic advantages over existing techniques. For example, as far as I am aware there is no description of the Illumina sequencing instrument recommended for QASeq or recommended coverage for accurate quantification. In relation to this, what is the approximate cost for each quantification experiment in comparison with comparable techniques?
2. Similarly, it would be interesting to provide more information on the maximum number of different modules that can be tested or what is the recommended number for accurate quantification. Also, is there any information on the performance of the different modules for the same target and advice in module design?
3. Given that tumors are intrinsically heterogeneous, what is the possible error in ploidy calculation associated to assuming that VAF in pure tumor is 50%? (line 364-365)

Minor comments:

4. I think it would be important to emphasize more the diagnostic advantages of QASeq in the discussion compared to other techniques.
5. Some parts of the text are a bit hard to follow. For example:
 - Line 255-256: Robust, consistent conversion yield and high number of quantitation modules are significant for QASeq.
 - Line 548-549: Number of amplicons = N means only using the N amplicons at 5' end of mRNA.
6. In general, gene names should be always in italics (including in figures) and numbers should be separated from units.
7. Why 175-plex QASeq is theoretically equivalent to 15225 different ddPCR CNV assays? (line 129-130). As far as I understand, only 175 targets are tested, which could correspond to 175 ddPCR independent

experiments (especially considering that some of these targets correspond to the reference controls).

8. Formulas in line 360-362 probably need some additional explanation (and would be more clearly expressed as a mathematical formula, not within the text).

9. Figure 2e. What do genes in gray mean?

10. Figure 4e legend: "...QASeq in four clinical FFPE CANCER SAMPLES and..."

Mario Cáceres

Reviewer #2 (Remarks to the Author):

The authors propose an alternative to droplet digital PCR (ddPCR) using next generation sequencing with a PCR-based molecular barcoding. Their approach is called quantitative amplicon sequencing (QASeq) for quantitation of specific DNA sequences. Whereas ddPCR has a limited number of fluorescence channels that can be used for quantitation that limits the sensitivity and precision, QASeq has the potential for more precise quantitation and sensitivity by using a large number of unique barcodes to distinguish between molecules. Results are presented for both 2-plex QASeq (two amplicon sequences) and for 175-plex QASeq panel capturing multiple CNV regions. The authors demonstrate a lower coefficient of variation and lower limit of detection compared to ddPCR. Applications include monitoring progression of metastatic disease in a small study of EGFR+HER2+ cancer patients, limit of detection for known copy number alterations, and quantitation of RNA abundance.

Barcoding to increase sensitivity and reduce false positives from sequencing error have been developed for use in conjunction with targeted next generation sequencing (Kinde et al., PNAS, 2011; Phallen et al, Science Translational Medicine, 2017). These existing methods primarily focused on somatic mutations. Does the novelty of QASeq primarily reside in the different applications for using barcoding with NGS -- in particular, CNV detection? An expanded discussion of how this technology differs from previous barcoding-based efforts for detection of tumor-derived molecules would be helpful.

Are there trade-offs to using a large number of UMIs such as sequencing errors and adapter dimers? Is there an effort to optimize the number of UMIs, or guidelines for this decision? Is the current approach to get around sequencing errors / primer dimer issues primarily a cutoff for the UMI's: "calculated as 5% of the mean of top 3 largest family size". How robust is quantitation of copy number, etc to the choice of cutoff?

How sensitive is copy number quantitation to the choice of a reference gene? Are differences in PCR efficiency due to GC content of the amplicons a source of variation? Are there opportunities for improving quantitation by modeling sources of PCR variation?

Can we consider QASeq comparable in both complexity and cost to ddPCR, or is there a large tradeoff in these parameters to obtain lower sensitivity and higher precision by QASeq?

Reviewer #3 (Remarks to the Author):

Wu and colleagues have developed an NGS method called QASeq based on PCR-mediated molecular barcoding and applied this towards CNV quantitation as well as RNA quantitation. The method takes advantage of increasing the number of loci (“modules”) being used to estimate the quantity as compared to digital PCR, which typically utilizes one or only a few targeted loci. Thus, QASeq is able to achieve accurate quantifications with lower variation as compared to ddPCR. The manuscript is generally clear and well written. The data presented appears generally sound and generally supports their conclusions, although there are some significant areas which should be addressed:

1. There are very many publications and methods and software tools for CNV estimation using NGS data as input. The authors do not demonstrate in a compelling way why their approach is a significant improvement over other methods for this purpose, for example DeviCNV, CovCopCan, TITAN, Sequenza, or CNVkit.
2. For the method to be replicated, much greater details in the Methods should be provided. Definitions should be provided for ‘fP’, ‘rPin’, and ‘rPout’ (e.g. which are used in Section 6 of the Supplementary Methods and Excel table), with an explanation for what is ‘H’ in the sequence, and which primers are the forward and reverse primers in the UMI PCR, nested PCR, and index PCR.
3. Raw ddPCR should also be provided in a public repository along with the NGS data to allow for replication.
4. The data from cell-free DNA is interesting, but what is missing is an estimation of what tumor fraction must be present in the sample for QASeq to be able to reliably call a copy number gain or loss.
5. When replicates are performed for QASeq or ddPCR, please indicate in the methods and supplementary methods to what degree they are truly independent replicates (same input, same day, same library prep, same sequencing run, etc?).
6. Small UMI family sizes are being removed (below 5% of the mean of the top 3). It is unclear why this threshold was chosen. How unstable are the CNV estimates when using all families, or when varying the cutoff for families to remove?
7. The experiments for ERBB2 CN utilize EIF2C1 as the reference, which is located on chromosome 1. This reference locus may not be appropriate, in particular for tumor specimens (e.g. Figure S3), where a number of tumor samples have a ploidy less than 2 (by both methods). It would be informative to analyze a series of breast tumors that are equivocal for HER2 status by a standard clinical test, e.g. single or dual-color ISH.
8. Following from the point above, the conclusion that QASeq could improve clinical sensitivity in CNV assessment (e.g. line 147; Figure S7) should also be put into context of specificity. Identifying more samples with ‘abnormal’ CN may not be clinically useful if there is an increase in false-positives.
9. The term “conversion yield” is a bit of a misnomer, as a significant part of the lowered yield seen in the ddPCR is due to dead volume space of the ddPCR instrument used. “Conversion” implies something to do with the chemistry being better, when the issue is more of hardware. Other dPCR instruments can have much lower dead volumes, for example the instrument from Combinati/ThermoFisher.
10. Newer dPCR instruments have up to 6 color channels, which allow for much greater capacity for multiplexing. With careful placement of signals in multidimensional space, it is possible to multiplex

much higher than simply the number of color channels. With greater multiplexing (multiple regions per gene for multiple genes), multi-channel dPCR CNV estimation should be able to match the “2-plex” QASeq data shown.

11. The sentence starting with “The physical...” on line 52 is unclear and should be revised.

12. For the 3 samples that were ddPCR positive by ERBB2 normal by QASeq and IHC (Figure 2d), what was the ddPCR ERBB2 CN? Were they just over the threshold for CN gain?

Reviewer #1 (Remarks to the Author):

The manuscript by Wu et al. describes an interesting new method for simultaneous absolute quantification of a certain number of targeted molecules based in next-generation sequencing. The authors describe the accuracy of the method compared to other techniques in different applications, such as estimation of CNV ploidy in tumor samples or cell free DNA and RNA quantification. They also show the application of the technique to different type of tumor samples and make a preliminary study of the association of disease progression and ERBB1 copy number in a small number of individuals. Overall the technique described appears to be reliable and accurate, improving some of the limitations of previous methods, and it is already included in the diagnostic company NUPROBE portfolio. However, so far the new biological insight gained is limited.

Major comments:

1. Given that the main contribution is the new method description, I think it would be useful for potential users to provide more practical information on its performance and diagnostic advantages over existing techniques. For example, as far as I am aware there is no description of the Illumina sequencing instrument recommended for QASeq or recommended coverage for accurate quantification. In relation to this, what is the approximate cost for each quantification experiment in comparison with comparable techniques?

We thank the reviewer for the comments and suggestions.

QASeq libraries were sequenced using Illumina HiSeq and NextSeq instruments in this study. The following description was included in the Methods Section: "*Sequencing was performed on HiSeq or NextSeq (Illumina) with 2X 150 bp paired-end reads and dual 8 bp index.*" Other information including recommended sequencing depth and comparison with other techniques were added in the manuscript and supporting information.

Recommended sequencing depth:

The recommended sequencing depth is 90,000X with 8.3 ng human DNA input (approximately 2,500 haploid copies) in multiplexed QASeq. At 90,000X depth, 20 M reads is suggested for the 223-module QASeq breast cancer liquid biopsy panel when 8.3 ng input is used. The sequencing depth should be adjusted proportionally to the input DNA amount, so that observed molecule count is not reduced due to insufficient reads.

The relationship between observed molecule count sequencing reads is illustrated in Figure S16. According to the recommendation of 90,000X depth with 8.3 ng DNA input, 108,000X depth should be used with 10 ng input, which corresponds to 19 M reads for 175-plex panel. This estimation is consistent with our observation that observed molecule counts reached plateau around 20 M reads.

Figure S16. Observed molecule counts at different reads. From a 31.6M read FASTQ file for 175-plex QASeq panel at 10 ng gDNA input, the sequencing file was down-sampled to different subfile sizes. The median molecule counts for the 175 modules were summarized under different reads. The molecule counts firstly increase as more reads are assigned and then reach a stable plateau.

We have added the above discussion and **Supplementary Fig. S16** in the Methods Section and Supplementary Section 6.

Comparison of QASeq with other techniques:

We have added Table S3 in the supplementary information to compare QASeq with whole exome sequencing and ddPCR in terms of cost, sample preparation, quantitation coverage range, and LOD for CNV/mutation.

Table S3. Comparison of QASeq with WES and ddPCR

	Cost	Sample preparation time	Quantitation Coverage	CNV LoD	Mutation LoD	Readout
QASeq	\$30 - \$250*	6 hours	1-223 regions	2.05 ploidy	0.1%	Sequencer
WES	~ \$500	1-2 days	Whole exome (semi-quantitative)	~2.4 ploidy	2%	
ddPCR	~ \$30	4 hours	1-6 regions	~2.4 ploidy	0.1% or lower	Droplet Reader

*QASeq cost varies based on the number of quantitation modules in a panel and varies using different sequencing instrument. 20 M reads is suggested for the QASeq breast cancer liquid biopsy panel containing 223 modules. The sequencing cost using NextSeq 550 high output cartridge is about \$200/sample. The sequencing cost will be significantly reduced to < \$50/sample if Novaseq 6000 or HiSeq X system is used.

2. Similarly, it would be interesting to provide more information on the maximum number of different modules that can be tested or what is the recommended number for accurate quantification. Also, is there any information on the performance of the different modules for the same target and advice in module design?

We thank the reviewer for the comments and suggestions. The largest QASeq panel tested in this study was the 223-module QASeq breast cancer liquid biopsy panel. By combining Simulated Annealing Design using Dimer Likelihood Estimation (SADDLE, Reference 31 in the manuscript), a primer set optimization software developed in our lab, with a nested protocol, about 60% on-target rate is maintained at 223-plex.

We have added discussion about QASeq maximum number of different modules in the manuscript Discussion section:

"A 384-plex standard PCR panel showed >40% on-target rate after optimized by SADDLE algorithm. Since a nested design is performed to further reduce non-specific amplification and primer dimer formation in QASeq protocol, we envision over 1000 modules can be simultaneously tested by QASeq."

We have added discussion about recommended module number based on desired CNV LoD in the Discussion section:

"The recommended number of modules in gene of interest is dependent on the desired limit of detection for copy number variation detection. As shown in Fig. 2a, though stochastic error in copy number quantitation was reduced by increasing the number of quantitation modules in the gene, there is a diminishing marginal utility as the module number increases. As a reference for roughly estimating the number of modules for different LoD requirement, CNV LoD of different genes with different module numbers per gene in the 175-plex QASeq panel was summarized (Supplementary Table S2)."

We have added Fig. S20 to illustrate the consistent performance of the different modules in the same target gene. Moreover, sub-gene level copy number variation can be detected if some modules are further changed compared to the rest of modules.

Figure S20. Representative performance of different modules in the same target gene from (a) gDNA from a healthy donor PBMC; (b,c) gDNA from fresh/frozen tissue of two breast cancer patients. Modules in *ERBB2* gene are highlighted. The ploidy values calculated from 49 different modules in *ERBB2* are highly consistent as shown in (a) and (b). Moreover, sub-gene level copy number variation is detected in (c) since the first five modules in *ERBB2* region are further amplified compared to the rest of *ERBB2* modules. Here modules are sorted based on chromosome location.

We have also added QASeq module design Sections in Supplementary Section 6:

“The general design workflow consists of five steps:

- 1) Deciding the number of modules in gene of interest. The recommended number of modules in gene of interest is dependent on the desired limit of detection for copy number variation detection. As a reference for roughly estimating the number of modules for different LoD requirement, CNV LoD of different genes with different module numbers per gene in the 175-plex QASeq panel, was summarized Supplementary Table S2 was summarized based on the performance of 175-plex QASeq panel and provided a reference for rough estimation of the number of modules for different LoD requirement.
- 2) Generate multiple primer candidates of forward primer (fP) and inner reverse primer (rPin) for each QASeq module. Genome context sequences based on regions of interest is downloaded. A single genome context sequence could have m fP candidates and n rPin candidates, thus combined into $m \times n$ primer pairs. Those primer pairs that satisfy specific amplicon length were selected as primer pair candidates for one module.
- 3) Optimize primer set of fP and rPin to minimize primer dimers for the whole panel, based on Simulated Annealing Design using Dimer Likelihood Estimation (SADDLE, Reference 31 in the manuscript), a primer set optimization software developed in our lab.
- 4) Based on the optimized fPs and rPins, generate multiple primer candidates for outer reverse primer (rPout). Candidates of rPout were generated so that the insert (the sequence between two primers) of fP and rPout was at least 4 nucleotides longer than the insert of fP and rPin for a nested design, to further reduce dimer or non-specific amplification.
- 5) Optimize primer set of fP and rPout to minimize primer dimers based on Simulated Annealing Design using Dimer Likelihood Estimation as previously mentioned.”

3. Given that tumors are intrinsically heterogeneous, what is the possible error in ploidy calculation associated to assuming that VAF in pure tumor is 50%? (line 364-365)

We thank the reviewer for the comments and suggestions. Considering the complexity of tumor heterogeneity, here we only proof-of-concept demonstrate the feasibility of inferring tumor *ERBB2* ploidy from plasma, with two assumption being made: (1) there is only one subclone in the tumor or

we take the average of the tumor tissue to treat tumor as a whole, and (2) the VAF in that pure tumor is 50% (monoallelic).

In more complex settings without the two assumptions, the ploidy and mutation VAF should be calculated as:

$$Ploidy(cfDNA) = \sum_1^i Ploidy(tumor\ clone\ i) \times Fraction(tumor\ clone\ i) + 2.0 \times (1 - \sum_1^i Fraction(tumor\ clone\ i)) \quad (1)$$

$$Mutation\ VAF(cfDNA) = \sum_1^i VAF(tumor\ clone\ i) \times Fraction(tumor\ clone\ i) \quad (2)$$

Here $Fraction(tumor\ clone\ i)$ is the fraction of circulation DNA derived from tumor subclone i in cfDNA, $Ploidy(tumor\ clone\ i)$ is the *ERBB2* ploidy in pure tumor clone i , and $VAF(tumor\ clone\ i)$ is the mutation VAF in pure tumor clone i .

Hypotheses need to be made to infer tumor *ERBB2* ploidy under limited information. When there is only one clone or we take the average of the tumor, equations are converted to:

$$Ploidy(cfDNA) = Ploidy(tumor) \times Fraction(tumor) + 2.0 \times (1 - Fraction(tumor)) \quad (3)$$

$$Mutation\ VAF(cfDNA) = VAF(tumor) \times Fraction(tumor) \quad (4)$$

Here we hypothesized mutation VAF in pure tumor is 50% (monoallelic), so that:

$$Fraction(tumor) = Mutation\ VAF(cfDNA) \times 2 \quad (5)$$

Therefore, possible error in the estimation of tumor *ERBB2* ploidy can come from both of the assumptions. If the VAF in pure tumor is 100% (biallelic), tumor fraction calculated from equation (5) is 2-fold of the true value. Overall, we envision the inferred mean tumor *ERBB2* ploidy based on equation (3) and (5) should be within two-fold of the true value to help estimate tumor tissue information from plasma cfDNA.

The discussion and updated calculation equations have been added in the Methods Section under "Infer tumor *ERBB2* ploidy from plasma" subsection.

Minor comments:

4. I think it would be important to emphasize more the diagnostic advantages of QASeq in the discussion compared to other techniques.

We thank the reviewer for the comments and suggestions.

We have added Supplementary Table S3 to compare QASeq with whole exome sequencing and ddPCR in terms of cost, sample preparation, quantitation coverage range, and LOD for CNV/mutation.

We added comparison of QASeq and ddPCR in the supplementary information Section 6:

"Low-plex QASeq DNA absolute quantitation modules showed comparable performance to ddPCR. With the scalability to highly multiplexed panels, QASeq improved CNV detection limit to below 2.05 ploidy. Furthermore, both CNV and mutation information are simultaneously provided from the NGS-based QASeq modules whereas ddPCR probes are designed for either CNV or mutation detection in one experiment."

We have also compared QASeq with other techniques in liquid biopsy setting in the Discussion section: *"The most accessible sample type for cancer monitoring is cfDNA derived from plasma. However, CNV monitoring using cfDNA is understudied comparing to mutation detection. Common CNV detection methods Immunohistochemistry and in situ hybridization are limited to tissue/cell samples, and not applicable to non-invasive cfDNA. Other cfDNA-compatible methods including ddPCR, NGS panels such as FoundationOne and Guardant 360, and microarray are not sensitive enough, all requiring > 25% heterozygous single copy loss or gain for detection which corresponds to 1.75 ploidy for loss and 2.25 ploidy for gain. The high CNV sensitivity of QASeq may allow better clinical sensitivity of copy number changes."*

5. Some parts of the text are a bit hard to follow. For example:

- Line 255-256: Robust, consistent conversion yield and high number of quantitation modules are significant for QASeq.

We thank the reviewer for the comments. We have modified the sentence to:

"Comparing to existing UMI-based methods which performed well on somatic mutation but were less sensitive to CNVs, the robust quantitation performance of QASeq is based on high and consistent conversion yield for each quantitation module, and the ability to scale up to high multiplexing."

- Line 548-549: Number of amplicons = N means only using the N amplicons at 5' end of mRNA.

We thank the reviewer for the comments. We have modified the sentence to:

"The standard deviation for relative expression level in triplicate experiments became lower as the number of amplicons per gene increased from 1 to 5. Only the amplicons at 5' end of mRNA were used when analyzing a subset of amplicons."

6. In general, gene names should be always in italics (including in figures) and numbers should be separated from units.

We thank the reviewer for pointing out the issue. We have edited gene names and numbers accordingly in the manuscript.

7. Why 175-plex QASeq is theoretically equivalent to 15225 different ddPCR CNV assays? (line 129-130). As far as I understand, only 175 targets are tested, which could correspond to 175 ddPCR independent experiments (especially considering that some of these targets correspond to the reference controls).

We thank the reviewer for the comments. Only one pair of targets are simultaneously analyzed in one ddPCR CNV assay. In QASeq, we can choose any 2 out of the 175 modules, the combination of which is equivalent to a ddPCR CNV assay. Therefore, the number of combinations can be mathematically expressed as $C(175,2) = 175 \times 174 \div 2 = 15225$.

In Supplementary Fig. S4, we showed ploidy value can be calculated from the UMI family counts of any 2 modules with one module set as target and the other one set as reference.

Figure S4. Pairwise ploidy analysis of 175-plex QASeq panel.

8. Formulas in line 360-362 probably need some additional explanation (and would be more clearly expressed as a mathematical formula, not within the text).

We thank the reviewer for the comments and suggestions. We have converted the formulas into mathematical equations (1)-(5) with explanation of each variable. We also added discussion about the assumptions being made and possible error for such estimation.

9. Figure 2e. What do genes in gray mean?

We thank the reviewer for the comments. The genes in gray have <3 modules per gene, and thus cannot be used to calculate gene ploidy using Mann-Whitney U test. We have updated the figure caption to add explanation.

10. Figure 4e legend: "...QASeq in four clinical FFPE CANCER SAMPLES and..."

We thank the reviewer for the comments. The figure 4e legend has been updated: "*Relative expression level measured by RNA QASeq in four clinical FFPE tumor tissue samples and 3 normal placenta FFPE samples.*"

Mario Cáceres

Reviewer #2 (Remarks to the Author):

The authors propose an alternative to droplet digital PCR (ddPCR) using next generation sequencing with a PCR-based molecular barcoding. Their approach is called quantitative amplicon sequencing (QASeq) for quantitation of specific DNA sequences. Whereas ddPCR has a limited number of fluorescence channels that can be used for quantitation that limits the sensitivity and precision, QASeq has the potential for more precise quantitation and sensitivity by using a large number of unique barcodes to distinguish between molecules. Results are presented for both 2-plex QASeq (two amplicon sequences) and for 175-plex QASeq panel capturing multiple CNV regions. The authors demonstrate a lower coefficient of variation and lower limit of detection compared to ddPCR. Applications include monitoring progression of metastatic disease in a small study of EGFR+HER2+ cancer patients, limit of detection for known copy number alterations, and quantitation of RNA abundance.

1. Barcoding to increase sensitivity and reduce false positives from sequencing error have been developed for use in conjunction with targeted next generation sequencing (Kinde et al., PNAS, 2011; Phallen et al, Science Translational Medicine, 2017). These existing methods primarily focused on somatic mutations. Does the novelty of QASeq primarily reside in the different applications for using barcoding with NGS -- in particular, CNV detection? An expanded discussion of how this technology differs from previous barcoding-based efforts for detection of tumor-derived molecules would be helpful.

We thank the reviewer for the comments and suggestions. We added Phallen et al, Science Translational Medicine, 2017 in the reference. Kinde et al., PNAS, 2011 was cited.

There are two differences that could contribute to QASeq CNV detection comparing to previous barcoding-based efforts which performed well on somatic mutation but were less sensitive to CNVs:

a. high and consistent conversion yield from run to run so that normalization from UMI count to ploidy is feasible. Here, conversion yield is the fraction of DNA molecules in a sample represented in the final NGS library.

The conversion yield performance is achieved by the PCR-based barcoding protocol as compared to ligation-based UMI attachment. Additionally, 30-min long annealing time is used to further improve the yield and reproducibility of PCR-based barcoding.

b. The ability to scale up to high multiplexing PCR so that the number of quantitation modules per gene can be adjusted, to overcome sampling error from Poisson distribution for CNV detection. The ability to scale up to high multiplexing, even when UMIs are incorporated, is enabled by combining Simulated Annealing Design using Dimer Likelihood Estimation (SADDLE), a primer set optimization software developed in our lab, with a nested protocol to further reduce primer dimer formation.

We have modified the Discussion section to better demonstrate the points:

"Comparing to existing UMI-based methods which performed well on somatic mutation but were less sensitive to CNVs, the robust quantitation performance of QASeq is based on high and consistent conversion yield for each quantitation module, and the ability to scale up to high multiplexing. We chose to perform two cycles of PCR-based barcoding with long annealing time, because the fraction of DNA molecules in a sample represented in the final NGS library is low for ligation-based UMI attachment. As the number of primer pairs increases for a multiplex PCR amplification, there is a combinatorial explosion of potential primer dimers and non-specific genomic amplification. The dimer problem is more complicated when UMIs must be incorporated in the multiplex PCR. By combining Simulated Annealing Design using Dimer Likelihood Estimation (SADDLE), a primer set optimization software developed in our lab, with a nested protocol, about 60% on-target rate is maintained even in the liquid biopsy panel with 223 modules."

2. Are there trade-offs to using a large number of UMIs such as sequencing errors and adapter dimers?

We thank the reviewer for the comments. The benefit of using a larger number of UMIs is that it will allow labeling of higher number of DNA molecules uniquely. On the other hand, longer UMIs may lead to higher chance of primer dimer formation and non-specific amplification in the genome.

Here we chose H_{15} to provide enough barcodes to label each DNA molecules uniquely for our experiment. The detailed optimization for composition and length of QASeq UMIs are demonstrated in the response for question 3 below. In current panels, the UMI chosen (H_{15}) does not cause much trouble in dimer formation or non-specific amplification. About 60% on-target rate is maintained even in the liquid biopsy panel with 223 modules.

We did not observe influence on sequencing error. UMI sequences with amplification or sequencing errors (i.e. G bases found in H_{15} UMIs which should not contain G; small UMI families that are likely results of UMI PCR / sequencing error) are discarded in analysis.

3. Is there an effort to optimize the number of UMIs, or guidelines for this decision?

We thank the reviewer for the comments. We have added the optimization for composition and length of UMIs in QASeq in Supplementary information Section 6:

UMI sequence. The degenerate base composition and length are optimized for QASeq panel. DNA sequences containing degenerate bases, such as poly(N) (i.e. mix of A, T, C, or G at each position), are often used as UMI sequences. In QASeq, we used poly(H) (A, T, or C) as UMI, because it has weaker cross-binding energy compared to poly(N) or mix of S (C or G) and W (A or T) bases as indicated by cross-binding energy calculation (Fig. S22).

The length of UMI determines how many molecules can be labeled uniquely. (H)₁₅ contains $3^{15} = 1.4 \times 10^7$ different sequences, which are enough for our planned molecule input. If 5,000 strands are used as input, H_{15} will allow 99.98% molecules to have unique UMI, and only 0.02% molecules may experience UMI collision by simulation. Even for 58,000 strands input (about 100 ng human gDNA), H_{15} will allow 99.6% molecules to have unique UMI.

Figure S22. Simulation of UMI cross-binding energy. Using (H)₂₀ instead of (N)₂₀ or (SWW)₆SW as UMI sequences reduces the mean cross-binding energy, indicating fewer potential primer-primer interactions to form dimers. Here 500 simulations were performed for each UMI pattern; in each simulation, 2 sequences that are consistent with the pattern were randomly generated, and the cross-binding ΔG° between these sequences were calculated at 60 °C and 0.18 M K^+ .

4. Is the current approach to get around sequencing errors / primer dimer issues primarily a cutoff for the UMI's: "calculated as 5% of the mean of top 3 largest family size".

We thank the reviewer for the comments. Dimer issue is reduced by combining Simulated Annealing Design using Dimer Likelihood Estimation (SADDLE) primer design software with a nested protocol in QASeq. Sequencing error in the targeted amplicon region is correctly by UMI. Sequencing error in UMI sequences lead to large number of small family sizes and is mostly overcome by removing UMIs not

matching the H₁₅ design and by dynamic cutoff of UMI family size. The details about UMI dynamic cutoff is demonstrated in the response for question 5 below.

5. How robust is quantitation of copy number, etc to the choice of cutoff?

We thank the reviewer for the comments. UMI family size cutoff is essential for accurate and robust quantitation, because large number of UMI families with small UMI family size (< 3) were observed which could be results of polymerase and sequencing errors in the UMI sequence. Although we removed UMIs not matching the poly H (A,T,C, no G) UMI design, small families splitted from large families due to UMI mutations were not fully removed.

Different cutoffs were evaluated. The number of observed molecules will decrease as the family size cutoff increases. X% of the mean of top 3 largest family size were tested as the cutoff, where as X = 0 (no cutoff), 3, 5, 10, 15, 20, 25 and 30. The calculated *ERBB2* ploidy using 2-plex panel was summarized in Figure S23.

Figure S23. Calculated *ERBB2* ploidy with different UMI family size cutoff. The cutoff was set as X% of the mean of top 3 largest family sizes. 2-plex QASeq panel as shown in manuscript Fig. 1b was analyzed here.

We showed that cutoff is necessary to get correct ploidy around 2 in a normal sample. There is no significant influence when the cutoff is larger than 5% of the mean of top 3. Furthermore, we evaluated the robustness in five technical replicates and selected X = 5 which minimized the variation (CV) of CNV quantitation in technical replicates.

The above discussion and Figure S23 was added in Supplementary Section 6.

It should be noted that further optimization on the choice of UMI family size cutoff may further improve the accuracy and robustness for absolute quantitation as well as copy number calculation.

6. How sensitive is copy number quantitation to the choice of a reference gene?

We thank the reviewer for the comments. The choice of reference matters for copy number quantitation, especially in clinical samples where the copy number can deviate from normal value of 2 in many regions of the genome as in Figure 2e of manuscript, from a breast cancer tumor fresh/frozen tissue DNA:

Therefore, references were selected based on iterative Mann-Witney U test in QASeq analysis workflow, without assuming any gene to serve as reference by default (Figure S5). *ERBB2* ploidy was calculated as 5.3 here using QASeq. In comparison, if we treat Chr17 (except those in *ERBB2* region) as the reference, *ERBB2* ploidy will be mistakenly reported as 7.4 due to copy number loss of Chr17. Indeed, the ploidy for Chr17 using all other modules not in Chr17 as the reference is 1.5, which is lower than the normal ploidy 2.

7. Are differences in PCR efficiency due to GC content of the amplicons a source of variation? Are there opportunities for improving quantitation by modeling sources of PCR variation?

We thank the reviewer for the comments and suggestions. We tried to seek the correlation between PCR conversion yield and a couple properties of the amplicon, but did not observed a clear correlation with GC content:

Since our lab previously used machine learning-based methods to predict DNA hybridization kinetics (Zhang et al, Nature Chemistry, 2018) or NGS sequencing depth from DNA sequence (Zhang et al, Nature Communications, 2021), we may be able to predict QASeq overall efficiency (conversion yield) based on DNA sequence.

8. Can we consider QASeq comparable in both complexity and cost to ddPCR, or is there a large tradeoff in these parameters to obtain lower sensitivity and higher precision by QASeq?

We thank the reviewer for the comments. We believe QASeq is comparable to ddPCR in terms of complexity and cost especially when we simultaneously analyze multiple regions of interest, as a result of the low throughput of ddPCR and the convenience of multiplexing in QASeq. QASeq is superior to ddPCR in terms of coverage and CNV LoD. Additionally, the DNA input requirement for ddPCR is proportional to the number of quantitation modules when scaling up to high multiplexing while QASeq does not need additional input for high number of quantitation modules.

We have added Supplementary Table S3 to compare QASeq with whole exome sequencing and ddPCR in terms of cost, sample preparation, quantitation coverage range, and LOD for CNV/mutation.

Table S3. Comparison of QASeq with WES and ddPCR

	Cost	Sample preparation time	Quantitation Coverage	CNV LoD	Mutation LoD	Readout
QASeq	\$30 - \$250*	6 hours	1-223 regions	2.05 ploidy	0.1%	Sequencer
WES	~ \$500	1-2 days	Whole exome (semi-quantitative)	~2.4 ploidy	2%	
ddPCR	~ \$30	4 hours	1-6 regions	~2.4 ploidy	0.1% or lower	Droplet Reader

*QASeq cost varies based on the number of quantitation modules in a panel and varies using different sequencing instrument. 20 M reads is suggested for the QASeq breast cancer liquid biopsy panel containing 223 modules. The sequencing cost using NextSeq 550 high output cartridge is about \$200/sample. The sequencing cost will be significantly reduced to < \$50/sample if Novaseq 6000 or Hiseq X system is used.

Reviewer #3 (Remarks to the Author):

Wu and colleagues have developed an NGS method called QASeq based on PCR-mediated molecular barcoding and applied this towards CNV quantitation as well as RNA quantitation. The method takes advantage of increasing the number of loci ("modules") being used to estimate the quantity as compared to digital PCR, which typically utilizes one or only a few targeted loci. Thus, QASeq is able to achieve accurate quantifications with lower variation as compared to ddPCR. The manuscript is generally clear and well written. The data presented appears generally sound and generally supports their conclusions, although there are some significant areas which should be addressed:

1. There are very many publications and methods and software tools for CNV estimation using NGS data as input. The authors do not demonstrate in a compelling way why their approach is a significant improvement over other methods for this purpose, for example DevicNV, CovCopCan, TITAN, Sequenza, or CNVkit.

We thank the reviewer for the comments and suggestions. We added side-by-side comparison of QASeq with CovCopCan and CNVkit for *ERBB2* copy number analysis in one normal PBMC DNA sample (expected *ERBB2* ploidy = 2.00), two reference spike-in samples prepared by mixing the normal PBMC DNA sample with *ERBB2*-positive cell line (SK-BR-3) DNA (expected *ERBB2* ploidy = 2.05 and 2.20), and three clinical cfDNA samples from breast cancer patients.

CovCopCan is designed for targeted sequencing, thus the analysis was performed using QASeq targeted sequencing data without considering UMI. Since CNVkit analysis is only compatible with whole exome sequencing (WES) or whole genome sequencing (WGS), the six selected samples were also sent for WES at Yale Center for Genome Analysis (YCGA). CNVkit analysis was performed on the WES data with mean depth > 150X for all samples. TITAN and Sequenza analysis were not compared in this study because they both require paired normal samples, which are not available. DevicNV and CovCopCan are both designed for CNV analysis using targeted sequencing data; we choose the newer method CovCopCan for comparison with QASeq here.

As summarized in Table S4, Fig. 2b and Figure S17-S19, QASeq was able to distinguish spike-in reference samples and clinical cfDNA samples with ploidy ≥ 2.05 from the normal sample. CovCopCan and CNVkit were not able to detect *ERBB2* CNV in the 2.05 or 2.20 ploidy reference samples. The cfDNA sample with *ERBB2* ploidy of 2.94 was called as *ERBB2* positive by all three methods; the sample with ploidy of 2.32 was identified by QASeq and CNVkit; the one with 2.17 ploidy was only identified by QASeq.

Based on these results, we believe QASeq has better CNV sensitivity than existing targeted sequencing-based or WES-based methods. Combining NGS-based accurate absolute quantitation module with high multiplexing to overcome molecule sampling stochasticity contributes to the improved CNV detection limit. We have added the comparison in Discussion Section of manuscript and Section 6 of Supporting Information.

Table S4. Comparison of QASeq with CovCopCan and CNVkit.

	Targeted amplicon sequencing		Whole exome sequencing	Sample notes
Sample	QASeq	CovCopCan	CNVkit	
Normal DNA	No CNV detected for ERBB2	No CNV detected for ERBB2	No CNV detected for ERBB2	Expected ploidy 2.00
Spike-in	2.06			Expected ploidy 2.05
	2.28			Expected ploidy 2.20
Clinical cfDNA 3679	2.17		2.72 - 2.78	Case 3368 time point 2
Clinical cfDNA 3669	2.32			Case 3669 time point 1
Clinical cfDNA 3934	3.94	3.52	3.69 - 3.76	Case 3669 time point 2

a. Normal sample

b. *ERBB2* Spike-in reference sample (Ploidy = 2.05)

c. *ERBB2* Spike-in reference sample (Ploidy = 2.20)

d. Clinical cfDNA sample (s-3679)

e. Clinical cfDNA sample (s-3669)

f. Clinical cfDNA sample (s-3934)

Figure S17. CovCopCan analysis of spike-in reference sample and clinical samples. Copy ratio plot was shown for normal sample (a), spike-in reference sample with expected *ERBB2* ploidy of 2.05 (b), spike-in reference sample with expected *ERBB2* ploidy of 2.20 (c), clinical sample S-3679 (d), clinical sample S-3669 (e), and clinical sample S-3934 (f).

Figure S18. CNVKit analysis of spike-in sample WES data. Copy ratio scatter plot with zoom in for *ERBB2* region was shown for normal sample (a), spike-in reference sample with expected *ERBB2* ploidy of 2.05 (b), and spike-in reference sample with expected *ERBB2* ploidy of 2.20 (c).

Figure S19. CNVKit analysis of clinical cfDNA sample WES data. Copy ratio scatter plot with zoom in for *ERBB2* region was shown for clinical sample S-3679 (a), clinical sample S-3669 (b), and clinical sample S-3934 (c).

2. For the method to be replicated, much greater details in the Methods should be provided. Definitions should be provided for 'fP', 'rPin', and 'rPout' (e.g. which are used in Section 6 of the Supplementary Methods and Excel table), with an explanation for what is 'H' in the sequence, and which primers are the forward and reverse primers in the UMI PCR, nested PCR, and index PCR.

We thank the reviewer for the comments and suggestions. We have added "Section 6. Supplementary Methods and Notes" in SI. Specifically, "QASeq panel design" was added which includes the definition of 'fP', 'rPin', and 'rPout' and the principles of how they are designed. Scheme of QASeq library preparation workflow was added as Figure S21. "UMI sequence" was added to demonstrate the definition of H (A, T, or C), the reason why we chose H as UMI, and the process we decide on length of UMI.

The added supporting information is also presented here:

QASeq panel design and scheme for library preparation workflow

The general design workflow consists of five steps:

1) Deciding the number of modules in gene of interest. The recommended number of modules in gene of interest is dependent on the desired limit of detection for copy number variation detection. As a reference for roughly estimating the number of modules for different LoD requirement, CNV LoD of different genes with different module numbers per gene in the 175-plex QASeq panel was summarized Supplementary Table S2 was summarized based on the performance of 175-plex QASeq panel and provided a reference for rough estimation of the number of modules for different LoD requirement.

2) Generate multiple primer candidates of forward primer (**fP**) and inner reverse primer (**rPin**) for each QASeq module. Genome context sequences based on regions of interest is downloaded. A single genome context sequence could have m fP candidates and n rPin candidates, thus combined into $m \times n$ primer pairs. Those primer pairs that satisfy specific amplicon length were selected as primer pair candidates for one module.

3) Optimize primer set of fP and rPin to minimize primer dimers for the whole panel, based on Simulated Annealing Design using Dimer Likelihood Estimation (SADDLE, Reference 31 in the manuscript), a primer set optimization software developed in our lab.

4) Based on the optimized fPs and rPins, generate multiple primer candidates for outer reverse primer (**rPout**). Candidates of rPout were generated so that the insert (the sequence between two primers) of fP and rPout was at least 4 nucleotides longer than the insert of fP and rPin for a nested design, to further reduce dimer or non-specific amplification.

5) Optimize primer set of fP and rPout to minimize primer dimers based on Simulated Annealing Design using Dimer Likelihood Estimation as previously mentioned.

Figure S21. Scheme of QASeq library preparation workflow

UMI sequence

The degenerate base composition and length are optimized for QASeq panel. DNA sequences containing degenerate bases, such as poly(N) (i.e. mix of A, T, C, or G at each position), are often used as UMI sequences. In QASeq, we used **poly(H) (A, T, or C) as UMI**, because it has weaker cross-binding energy compared to poly(N) or mix of S (C or G) and W (A or T) bases as indicated by cross-binding energy calculation (Fig. S22).

The length of UMI determines how many molecules can be labeled uniquely. **H₁₅** contains 1.4×10^7 different sequences, which are enough for our planned molecule input. If 5,000 strands are used as input, H₁₅ will allow 99.98% molecules to have unique UMI, and only 0.02% molecules may experience UMI collision by simulation. Even for 58,000 strands input (about 100 ng human gDNA), H₁₅ will allow 99.6% molecules to have unique UMI.

Figure S22. Simulation of UMI cross-binding energy. Using (H)₂₀ instead of (N)₂₀ or (SWW)₆SW as UMI sequences reduces the mean cross-binding energy, indicating fewer potential primer-primer interactions to form dimers. Here 500 simulations were performed for each UMI pattern; in each simulation, 2 sequences that are consistent with the pattern were randomly generated, and the cross-binding ΔG° between these sequences were calculated assuming 60 °C and 0.18 M K⁺.

3. Raw ddPCR should also be provided in a public repository along with the NGS data to allow for replication.

We thank the reviewer for the comments and have uploaded all ddPCR raw data together with NGS data. We modified Data availability section accordingly.

4. The data from cell-free DNA is interesting, but what is missing is an estimation of what tumor fraction must be present in the sample for QASeq to be able to reliably call a copy number gain or loss.

We thank the reviewer for the comments. The tumor fraction that must be present in a sample for QASeq to call a CNV event is dependent on the ploidy of tumor tissue. With the ability to distinguish 2.05 ploidy from normal case, the lowest tumor fraction that can be detected is 0.5% assuming tumor gene ploidy is 12 (high amplification). The minimum tumor fraction will be reduced to 2.5% when tumor ploidy is 4.

The LoD for copy number loss is calculated to be 1.97 for *ERBB2* gene in Table S2. Based on this LoD, the lowest tumor fraction that can be detected is 3% assuming tumor gene ploidy is 1, in the case of heterozygous single copy loss.

We have added the above discussion in Section 6 of Supporting information.

5. When replicates are performed for QASeq or ddPCR, please indicate in the methods and

supplementary methods to what degree they are truly independent replicates (same input, same day, same library prep, same sequencing run, etc?).

We thank the reviewer for the comments. QASeq replicates were performed by the same operator, using the same DNA input, with library preparation performed in the same day and sequenced in the same run. The ddPCR replicates were performed by the same operator, using the same DNA input, with droplet generation and PCR reaction performed in the same day and analyzed by Droplet Reader in the same run.

We have added the above details about replicates in Methods Section.

6. Small UMI family sizes are being removed (below 5% of the mean of the top 3). It is unclear why this threshold was chosen. How unstable are the CNV estimates when using all families, or when varying the cutoff for families to remove?

We thank the reviewer for the comments. UMI family size cutoff is essential for accurate and robust quantitation, because large number of UMI families with small UMI family size (< 3) were observed which could be results of polymerase and sequencing errors in the UMI sequence. Although we removed UMIs not matching the poly H (A,T,C, no G) UMI design, small families splitted from large families due to UMI mutations were not fully removed.

Different cutoffs were evaluated. The number of observed molecules will decrease as the family size cutoff increases. X% of the mean of top 3 largest family size were tested as the cutoff, where as $X = 0$ (no cutoff), 3, 5, 10, 15, 20, 25 and 30. The calculated *ERBB2* ploidy using 2-plex panel was summarized in Figure S23.

Figure S23. Calculated *ERBB2* ploidy with different UMI family size cutoff. The cutoff was set as X% of the mean of top 3 largest family sizes. 2-plex QASeq panel as shown in manuscript Fig. 1b was analyzed here.

We showed that cutoff is necessary to get correct ploidy around 2 in a normal sample. There is no significant influence when the cutoff is larger than 5% of the mean of top 3. Furthermore, we evaluated the robustness in five technical replicates and selected $X = 5$ which minimized the variation (CV) of CNV quantitation in technical replicates.

The above discussion and Figure S23 was added in Supplementary Section 6.

It should be noted that further optimization on the choice of UMI family size cutoff may further improve the accuracy and robustness for absolute quantitation as well as copy number calculation.

7. The experiments for *ERBB2* CN utilize *EIF2C1* as the reference, which is located on chromosome 1. This reference locus may not be appropriate, in particular for tumor specimens (e.g. Figure S3), where a number of tumor samples have a ploidy less than 2 (by both methods). It would be informative to analyze a series of breast tumors that are equivocal for HER2 status by a standard clinical test, e.g. single or dual-color ISH.

We thank the reviewer for the comments. We took *EIF2C1* as the reference in ddPCR experiments since *EIF2C1* reference assay is a common standard in ddPCR copy number analysis. 2-plex QASeq used *EIF2C1* as reference as well for side-by-side demonstration purpose (Fig. 1). QASeq assay that were applied to clinical samples did not use *EIF2C1* as the reference. Fresh/frozen tumor samples (those in Figure 2 and Figure S3) were assayed by QASeq 175-plex panel in which 49 modules were from *ERBB2* and the rest 126 modules were located on different chromosomes. References were

selected based on iterative Mann-Witney U test, without assuming any gene to serve as reference by default for all clinical samples.

We agree that using *EIF2C1* alone as the reference may be inappropriate for complex clinical samples. There were 3 samples with *ERBB2* ploidy > 2 in ddPCR but normal in QASeq and IHC (Fig. 2d), which may be due to the pitfalls of single reference *EIF2C1* in ddPCR assay. Since the fresh frozen tumor tissue samples were purchased from commercial vendor (OriGene Technologies) and *ERBB2* status of the tumor tissue were only measured by IHC, we did not have ploidy data from single or dual-color ISH test.

Furthermore, serial longitudinal plasma cfDNA samples were tested using QASeq breast cancer liquid biopsy panel, from which tumor *ERBB2* ploidy was inferred. The inferred tumor *ERBB2* ploidy by QASeq was consistent with the available tumor ploidy from FISH (Fig. 3d).

We did not have samples available with equivocal *ERBB2* status in this study. We would like to pursue deeper on the potential of QASeq technology in collaboration with clinicians in the next studies, and really appreciate the suggestion of including a series of breast tumors that are equivocal for HER2 status.

8. Following from the point above, the conclusion that QASeq could improve clinical sensitivity in CNV assessment (e.g. line 147; Figure S7) should also be put into context of specificity. Identifying more samples with 'abnormal' CN may not be clinically useful if there is an increase in false-positives.

We thank the reviewer for the comments. Though QASeq analytical accuracy of copy number calculation was validated by spike-in samples and comparison with other methods, indeed the clinical significance of 'abnormal' copy number is not well-established especially for small changes.

Nevertheless, with the new tools including QASeq developed, which are capable of detecting small copy number change while applicable to cell-free DNA, the technical gap for sensitive detection of 'abnormal' copy number and investigating the clinical significance is being filled. We will further follow up on the clinical significance and will be delighted to see other researchers/clinicians use the new tools as well.

9. The term "conversion yield" is a bit of a misnomer, as a significant part of the lowered yield seen in the ddPCR is due to dead volume space of the ddPCR instrument used. "Conversion" implies something to do with the chemistry being better, when the issue is more of hardware. Other dPCR instruments can have much lower dead volumes, for example the instrument from Combinati/ThermoFisher.

We thank the reviewer for the comments. Conversion yield is defined as the fraction of input molecules that are observed in the experiment. This parameter reflects the overall performance in all steps: in QASeq it considers PCR barcoding yield and loss in bioinformatics; in ddPCR it includes droplet generation and dead volume loss. QASeq showed good conversion yield (86% on average) in the 2-plex demonstration.

Indeed the dead volume loss is instrument-specific, and conversion yield could be high in other ddPCR instruments. We have modified first paragraph of discussion section in the manuscript accordingly: *"Advances have been made to digital PCR systems in terms of both multiplexing and dead volume. As an alternative approach to ddPCR for nucleic acid quantitation, the multiplexing ability for QASeq is still one order of magnitude higher than the best digital PCR leading to better CNV LoD using highly multiplexed QASeq panel."*

10. Newer dPCR instruments have up to 6 color channels, which allow for much greater capacity for multiplexing. With careful placement of signals in multidimensional space, it is possible to multiplex much higher than simply the number of color channels. With greater multiplexing (multiple regions per gene for multiple genes), multi-channel dPCR CNV estimation should be able to match the "2-plex" QASeq data shown.

We thank the reviewer for the comments and pointing out the latest technical advances. We have added academic references (Anal. Chem. 2021, 93, 10538–10545; Oncotarget, 2018, 9, 37393-37406) in Discussion Section to acknowledgement the development of new six-color instruments. We are also glad to see proof of concept SAGAsafe® 24-plex assay using Stilla’s six-color naica® system in 2021 ASHG meeting after initial submission of this manuscript. Better multi-channel digital PCR will definitely improve copy number quantitation accuracy.

As an alternative approach to ddPCR for absolute quantitation and CNV quantitation, the multiplex ability for QASeq is still over one order of magnitude higher than the best digital PCR leading to better CNV LoD using highly multiplexed QASeq panel. Since the current on-target rate of 223-plex QASeq is high, we believe that QASeq can be expanded to >1000-plex without too much optimization. Furthermore, both CNV and mutation information are simultaneously provided from the NGS-based QASeq modules, whereas ddPCR probes are designed for either CNV or mutation detection in one experiment. We have modified the first paragraph of discussion section in the manuscript and Supplementary Section 6 in SI.

11. The sentence starting with “The physical...” on line 52 is unclear and should be revised.

We thank the reviewer for the comments. The sentence has been revised to:

“The intrinsic limitation due to stochasticity in molecule sampling process leads to the observed number of DNA molecules, and thus the observed ploidy, deviating from the expected “true value” in CNV quantitation (Supplementary Fig. S1).”

12. For the 3 samples that were ddPCR positive by *ERBB2* normal by QASeq and IHC (Figure 2d), what was the ddPCR *ERBB2* CN? Were they just over the threshold for CN gain?

We thank the reviewer for the comments. Yes, the ploidy for all the three samples were below or around 3. The ddPCR *ERBB2* copy number for the three samples were 2.50, 2.58 and 3.01 respectively.

Please find below the source data for Fig. 2d. All source data for each figure were provided along with the manuscript.

Sample	QASeq	ddPCR			IHC	QASeq	ddPCR
	Ploidy	Ploidy rep1	Ploidy rep2	Mean		Status (pos=1, neg=0)	
FF1	7.63	7.10	6.83	6.97	1	1	1
FF2	11.83	12.31	10.98	11.64	1	1	1
FF3	1.89	2.00	2.07	2.03	0	0	0
FF4	1.93	1.84		1.84	1	0	0
FF5	1.85	1.79	1.76	1.77	0	0	0
FF6	13.66	13.70		13.70	1	1	1
FF7	7.74	7.86		7.86	1	1	1
FF8	2.00	1.54		1.54	0	0	0
FF9	1.83	1.98		1.98	0	0	0
FF10	1.48	1.61		1.61	0	0	0
FF12	2.00	1.41		1.41	0	0	0
FF13	2.00	3.01		3.01	0	0	1
FF14	34.76	36.93		36.93	1	1	1
FF15	6.92	8.18		8.18	1	1	1
FF16	2.00	2.50		2.50	0	0	1
FF17	1.37	1.76		1.76	0	0	0
FF18	1.48	2.58		2.58	0	0	1
FF19	5.29	5.07		5.07	1	1	1

REVIEWERS' COMMENTS

Reviewer #1 (Remarks to the Author):

The authors have made a considerable effort to answer the reviewer concerns and all my comments have been addressed satisfactorily.

Mario Cáceres

Reviewer #2 (Remarks to the Author):

The authors have addressed each of my concerns.

Reviewer #3 (Remarks to the Author):

Wu and colleagues have developed an NGS method called QASeq based on PCR-mediated molecular barcoding and applied this towards CNV quantitation as well as RNA quantitation. The method takes advantage of increasing the number of loci ("modules") being used to estimate the quantity as compared to digital PCR, which typically utilizes one or only a few targeted loci. Thus, QASeq is able to achieve accurate quantifications with lower variation as compared to ddPCR. The manuscript is generally clear and well written. The data presented appears generally sound and generally supports their conclusions.

This reviewer appreciates the authors' comprehensive responses to each of the reviewers' comments and questions. The main concerns have been addressed adequately and I have no further critiques.